# Inner membrane complex proteomics reveals a palmitoylation regulation critical for intraerythrocytic development of malaria parasite

Pengge Qian[1†], Xu Wang[1†], Chuan-Qi Zhong[1†], Jiaxu Wang[2†], Mengya Cai[1], Wang Nguitragool[3], Jian Li[1*], Huiting Cui[1*], Jing Yuan[1*]

[1]State Key Laboratory of Cellular Stress Biology, School of Life Sciences, Faculty of Medicine and Life Sciences, Xiamen University, Xiamen, China; [2]Xiamen Center for Disease Control and Prevention, Xiamen, China; [3]Department of Molecular Tropical Medicine and Genetics, Faculty of Tropical Medicine, Mahidol University, Ratchathewi, Thailand

*For correspondence:
jianli_204@xmu.edu.cn (JL);
cuihuiting@xmu.edu.cn (HC);
yuanjing@xmu.edu.cn (JY)

[†]These authors contributed equally to this work

**Competing interest:** The authors declare that no competing interests exist.

**Abstract** Malaria is caused by infection of the erythrocytes by the parasites *Plasmodium*. Inside the erythrocytes, the parasites multiply via schizogony, an unconventional cell division mode. The inner membrane complex (IMC), an organelle located beneath the parasite plasma membrane, serving as the platform for protein anchorage, is essential for schizogony. So far, the complete repertoire of IMC proteins and their localization determinants remain unclear. Here we used biotin ligase (TurboID)-based proximity labeling to compile the proteome of the schizont IMC of the rodent malaria parasite *Plasmodium yoelii*. In total, 300 TurboID-interacting proteins were identified. 18 of 21 selected candidates were confirmed to localize in the IMC, indicating good reliability. In light of the existing palmitome of *Plasmodium falciparum*, 83 proteins of the *P. yoelii* IMC proteome are potentially palmitoylated. We further identified DHHC2 as the major resident palmitoyl-acyl-transferase of the IMC. Depletion of DHHC2 led to defective schizont segmentation and growth arrest both in vitro and in vivo. DHHC2 was found to palmitoylate two critical IMC proteins CDPK1 and GAP45 for their IMC localization. In summary, this study reports an inventory of new IMC proteins and demonstrates a central role of DHHC2 in governing the IMC localization of proteins during the schizont development.

## Editor's evaluation

Apicomplexan parasites, including the malaria parasite Plasmodium, possess a characteristic inner membrane complex (IMC), which plays an essential role in maintaining the parasite shape and regulating motility. In this study, the authors used proximity labeling to determine the IMC proteome of erythrocytic stages in the rodent malaria parasite Plasmodium yoelii. They identify the palmitoyl-acyl-transferase DHHC2 as a key enzyme that regulates the localization of IMC proteins through palmitoylation. This work provides new insights into the function of the IMC in the erythrocytic stages of the malaria parasite.

## Introduction

Malaria, a human scourge caused by the protozoans of the genus *Plasmodium*, affects more than 200 million people and is responsible for approximately half a million deaths in 2019 (***World Health***

*Organization, 2019*). The symptoms of malaria are caused by the intraerythrocytic proliferation of the parasites. After erythrocyte invasion, the parasites replicate via schizogony to produce up to 32 invasive daughter cells called the merozoites. Following their release from the host cell, these merozoites invade other erythrocytes and continue the intraerythrocytic life cycle.

A defining feature of apicomplexan parasites, including the *Plasmodium,* is the organelle inner membrane complex (IMC) which are flattened membranous vesicles beneath the plasma membrane (PM). In *Plasmodium*, the IMC is present in the invasive or motile stages such as the merozoites, ookinetes, and sporozoites of all *Plasmodium*, as well as in the male gametocytes of human pathogen *P. falciparum* (*Morrissette and Sibley, 2002*; *Kono et al., 2012*). In the asexual cycle of the *Plasmodium*, the biogenesis of the IMC begins at the early schizont stage after organelle duplication in the cytoplasm and several rounds of genome replication within the intact nucleus (*Matthews et al., 2018*). Allocation and packaging of nuclei and organelles into daughter cells are achieved by cytokinesis, in which the PM and the extending IMC coordinately invaginate and surround each daughter haploid merozoite (*Ferreira et al., 2020*). As a result, the pellicle of merozoite is composed of a single membrane of PM, closely aligned double membranes of the IMC, and 2–3 subpellicular microtubules (SPMT) associated with the IMC (*Kono et al., 2016*). After invasion into a new erythrocyte, the IMC dissembles at the early ring stage (*Ferreira et al., 2020*). The IMC is best recognized as the platform for attaching the actomyosin motor complex-glideosome (occupying the space between PM and IMC) responsible for parasite invasion and motility (*Harding and Frischknecht, 2020*; *Frénal and Soldati-Favre, 2013*). It also serves to maintain the cell shape and rigidity of the merozoite (*Frénal et al., 2017*). The cytoplasmic side of IMC is associated with a rigid meshwork composed of several families of proteins, including alveolins (*Gould et al., 2008*), which is important for IMC-SPMT interconnection.

To understand the role(s) of the IMC in the parasite development, obtaining the complete list of protein components of the IMC is important. However, systematic proteomic analyses on the IMC protein composition have not been reported due to the difficulty in separating the IMC from other membranes, such as the PM. So far, a limited number of IMC and IMC-associated proteins have been identified, by either candidate gene strategy or isolating the interactors of IMC protein. Known IMC proteins include glideosome-associated protein 45 (GAP45) (*Perrin et al., 2018*), GAP40 (*Frénal et al., 2010*), GAP50 (*Bosch et al., 2012*), myosin A tail domain interacting protein (MTIP) (*Bergman et al., 2003*), MyoA (*Bergman et al., 2003*), ECL1 (*Green et al., 2017*), GAP with multiple-membrane spans 1 (GAPM1), GAPM2, GAPM3 (*Bullen et al., 2009*), ISP1, and ISP3 (*Wang et al., 2020*) as well as the alveolin proteins IMCp, IMC1c, IMC1e, IMC1f, and IMC1g (*Kono et al., 2012*). In addition, calcium-dependent kinase 1 (CDPK1) and photosensitized INA-labeled protein 1 (PhIL1) were also reported to reside in the IMC of schizonts (*Kumar et al., 2017*; *Saini et al., 2017*). BCP1, MORN1, and CINCH, components of the basal complex, a subcompartment of the IMC at the posterior extending edge, were also recently identified (*Morano and Dvorin, 2021*). Remarkably, most of the known IMC proteins were refractory to gene deletion, suggesting an essential function in the development of asexual blood stage (*Ferreira et al., 2020*). Among the IMC proteins, GAP45 and CDPK1 have been studied extensively. CDPK1 plays a key role in early schizont development as its depletion or inhibition causes parasite arrest at the early schizonts (; *Kumar et al., 2017*; *Azevedo et al., 2013*). On the other hand, GAP45 is essential for merozoite invasion (*Perrin et al., 2018*). Despite great efforts in IMC research, many questions still remain regarding the IMC components and their function in schizogony. How many proteins are there in IMC? What determines IMC protein localization?

Enzyme-catalyzed proximity labeling (PL) coupled with mass spectrometry (MS) offers an alternative approach for proteome discovery (*Kim and Roux, 2016*). BioID is an engineered bacterial biotin ligase adapted for proximity-based biotinylation of proteins in living cells (*Roux et al., 2012*). The proteins covalently labeled with biotin can be isolated by streptavidin-biotin affinity purification followed by MS analysis. BioID has been used to identify the protein components of complexes and organelles in different model organisms (*Qin et al., 2021*). In *Plasmodium*, BioID-based PL has also generated organelle- and vesicle-specific proteins or proteomes, including those of the gametocyte-specific osmiophilic bodies of *P. berghei*, the blood stage parasitophorous vacuolar membrane of *P. berghei* and *P. falciparum*, the apicoplast of *P. falciparum*, and the IMC and apical annuli proteins of *P. falciparum* (*Geiger et al., 2020*; *Boucher et al., 2018*; *Schnider et al., 2018*; *Khosh-Naucke et al., 2018*; *Kehrer et al., 2016*; *Wichers et al., 2021*). These analyses have led to the functional discovery of previously undescribed proteins, providing new insights in the organelle biology of

malaria parasites. However, BioID requires parasite exposure to biotin over a long period (18–24 hr), which is not ideal or feasible for certain developmental stages with a short life span. In addition, BioID does not work well at temperatures below 37°C (*May et al., 2020*; *Bosch et al., 2021*), rendering its application to the mosquito stages of *Plasmodium* unsuitable. Recently, a new biotin ligase, TurboID, was developed by directed evolution (*Branon et al., 2018*). Compared to BioID, TurboID is faster and can work under a broader range of temperatures (*Branon et al., 2018*). So far, the application of TurboID in the *Plasmodium* has not been reported.

In this study, we applied TurboID-PL and quantitative MS to obtain a proteome of the IMC in the schizonts of rodent malaria parasite *P. yoelii*. IMC targeting was achieved by fusing TurboID with the N-terminal 20 residues of ISP1, a known IMC resident protein. A collection of 300 proteins were identified as candidate IMC and IMC-associated proteins, of which 83 are potentially palmitoylated. We further demonstrated DHHC2 as a master IMC palmitoyl-acyl-transferase which plays a critical role in schizont segmentation and merozoite invasion by regulating the IMC localization of CDPK1 and GAP45 via palmitoylation.

## Results

### Biotin-labeling of *Plasmodium* proteins by TurboID ligase

To test the activity of the TurboID ligase relative to the BioID ligase for PL of malaria parasites, we fused a hemagglutinin (HA) tag to the N-terminus of each ligase (*Figure 1—figure supplement 1A*). These ligases were episomally expressed in the asexual blood stages of *P. yoelii* under the promoter of the *isp3* gene (*Figure 1—figure supplement 1A*), a gene that is highly transcribed in the schizonts (*Otto et al., 2014*). Immunoblot detected comparable BioID and TurboID expressions in the asexual blood stages (*Figure 1—figure supplement 1B*). Different from an automatous rupture of mature schizonts of the in vitro cultured *P. falciparum*, the *P. yoelii* schizonts displayed an arrest in rupture after maturation in the in vitro condition, which permits PL of mature schizonts. The schizonts expressing each ligase were incubated with 100 µM biotin at 37°C for different times (0.25, 1, 3, and 18 hr). Immunoblot using streptavidin-HRP detected robust protein biotinylation in the cell extracts of TurboID-parasites as early as 0.25 hr after biotin incubation (*Figure 1—figure supplement 1C*). In contrast, protein biotinylation in the BioID-parasites appeared at a low level at 3 hr and reached a high level at 18 hr (*Figure 1—figure supplement 1C*). To confirm these results, dot blot experiments were performed using streptavidin-HRP and similar results were observed (*Figure 1—figure supplement 1D*). Next, we tested the temperature compatibility of the two ligases for PL in the parasites. BioID- and TurboID-schizonts were incubated for 18 and 3 hr, respectively, with 100 µM biotin at different temperatures (4, 22, 30, and 37°C). Biotin-incubated parasites stained with fluorescently conjugated streptavidin revealed that both ligases had similar labeling activity at 30 and 37°C (*Figure 1—figure supplement 1E,F*). Notably, only TurboID retained its activity at 22°C (*Figure 1—figure supplement 1E,F*). We further performed dot blots using streptavidin-HRP and obtained similar results (*Figure 1—figure supplement 1G*). These results indicate that TurboID is active at temperatures lower than 37°C, a temperature required for BioID to be fully functional. We also tested TurboID-mediated PL in the ookinetes, a mosquito stage of parasites with a preferential living temperature at 22°C. Cultured ookinetes from the BioID- and TurboID-parasites were incubated for 18 and 3 hr, respectively, with 100 µM biotin at 22°C. Costaining with the fluorescently conjugated streptavidin and anti-HA antibody detected cytosolic and nuclear protein biotinylation only in the TurboID-ookinetes when exogenous biotin was added (*Figure 1—figure supplement 1H*). Compared with BioID, TurboID allowed more robust PL of proteins in the living parasites with a shorter biotin incubation time and is less temperature sensitive.

### Detection of IMC proteins using TurboID labeling and quantitative mass spectrometry

Next, we applied TurboID for PL of the IMC to identify new IMC proteins in the schizonts. The HA-tagged TurboID was fused with an IMC targeting peptide, the N-terminal 20 residues of ISP1 (Tb-IMC) (*Wang et al., 2020*; *Wetzel et al., 2015*; *Figure 1—figure supplement 2A*). The HA-tagged TurboID alone (Tb-cyto) served as a control to indicate non-specific biotinylation (*Figure 1A* and *Figure 1—figure supplement 2A*), permitting specific identification of IMC and IMC-associated

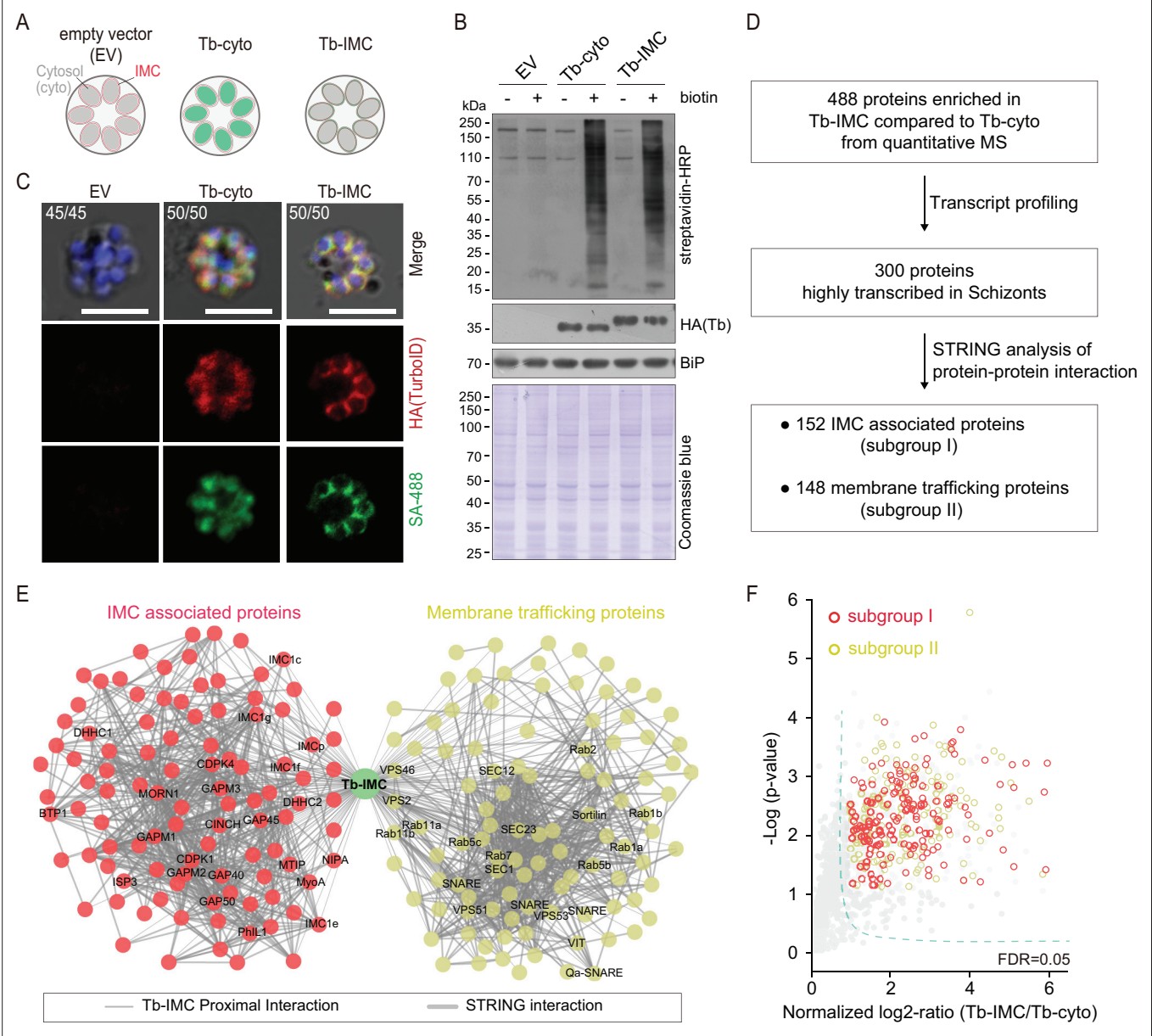

**Figure 1.** Proteomic of *P. yoelii* schizont IMC by TurboID and quantitative mass spectrometry. (**A**) Schematic of schizonts with TurboID ligase localizing in the cytoplasm (Tb-cyto) and IMC (Tb-IMC). EV indicates the schizonts expressing empty vector (EV). See the detailed information of Tb-IMC and Tb-cyto in *Figure 1—figure supplement 2*. (**B**) Immunoblot and streptavidin blot of total lysate from the schizonts expressing the EV, Tb-cyto and Tb-IMC. Tagged ligase was detected by anti-HA antibody while biotinylated proteins were detected by streptavidin-conjugated horseradish peroxidase (streptavidin-HRP). Comparable loaded lysate was indicated by BiP control and Coomassie blue stain. (**C**) Costaining of TurboID ligase and biotinylated proteins in the schizonts expressing the ligase of the EV, Tb-cyto, and Tb-IMC. The schizonts incubated with or without 100 µM biotin were costained with the SA-488 and anti-HA antibody. x/y in the figure is the number of cells displaying signal/the number of cells analyzed. Scale bar = 5 µm. (**D**) Workflow for filtering the Tb-IMC interacting proteins (proximal interactors). 488 biotinylated proteins that were at least two times more abundant in Tb-IMC than that in Tb-cyto control among three replicates and with an adjusted p-value<0.05, were significantly enriched. Detailed information in *Figure 1—figure supplement 2*. (**E**) Interaction network of 300 Tb-IMC interacting proteins (STRING, p-value <1.0e-16, bold lines). Two subgroups (I: left, II: right) were functionally clustered. Many known inner membrane complex (IMC) or IMC-associated proteins were clustered into the subgroup I while many annotated ER/Golgi secretory or vesicle trafficking proteins were clustered into the subgroup II. (**F**) Volcano plots showing the 300 Tb-IMC interacting proteins. Relative biotinylation of each protein was calculated by quantifying protein intensity in Tb-IMC relative to Tb-cyto schizonts (n=3). Proteins in the subgroup I (red circle) and subgroup II (yellow circle) are indicated. MS: mass spectrometry; HA: hemagglutinin.

The online version of this article includes the following source data and figure supplement(s) for figure 1:

**Source data 1.** (related to *Figure 1B*) Immunoblot and streptavidin blot of total lysate from the schizonts expressing the EV, Tb-cyto and Tb-IMC.

*Figure 1 continued on next page*

*Figure 1 continued*

**Figure supplement 1.** Protein proximity labeling by biotin ligases BioID and TurboID in the *P. yoelii* parasites.

**Figure supplement 1—source data 1.** (related to *Figure 1—figure supplement 1B*) Immunoblot of the HA-tagged ligases episomally expressed in the asexual blood stages of the *P. yoelii*.

**Figure supplement 1—source data 2.** (related to *Figure 1—figure supplement 1C*) Streptavidin blot detecting the biotinylated proteins from the wildtype, BioID or TurboID-expressing parasites incubated with exogenous biotin for different times (0, 0.25, 1, 3, and 18 hr) at 37°C.

**Figure supplement 1—source data 3.** (related to *Figure 1—figure supplement 1D*) Streptavidin dot blot of the biotinylated proteins from total lysate of parasites in C.

**Figure supplement 1—source data 4.** (related to *Figure 1—figure supplement 1D*) Quantification of streptavidin dot blot of the biotinylated proteins.

**Figure supplement 1—source data 5.** (related to *Figure 1—figure supplement 1F*) Quantification of IFA signal intensity.

**Figure supplement 1—source data 6.** (related to *Figure 1—figure supplement 1G*) Streptavidin dot blot of the biotinylated proteins from the parasite lysates.

**Figure supplement 2.** TurboID-mediated labeling of IMC proteins in the schizonts.

**Figure supplement 2—source data 1.** (related to *Figure 1—figure supplement 2B*) Immunoblot of the HA-tagged ligase from the total lysate of schizonts expressing the Tb-cyto or Tb-IMC.

**Figure supplement 2—source data 2.** (related to *Figure 1—figure supplement 2D*) Streptavidin dot blot of biotinylated proteins in the EV, Tb-cyto, Tb-IMC expressing schizonts incubated with or without 100 μM biotin.

**Figure supplement 2—source data 3.** (related to *Figure 1—figure supplement 2F*) Immunoblot of streptavidin-affinity purified biotinylated proteins in Tb-IMC schizonts.

**Figure supplement 3.** Predicated functional profile of the Tb-IMC interacting proteins Gene ontology analysis of the 300 Tb-IMC interacting proteins.

proteins. Both ligases (Tb-IMC and Tb-cyto) were driven by the promoter of gene *isp3* and episomally expressed in the asexual blood stages (*Figure 1—figure supplement 2B*). As expected, the Tb-IMC ligase dominantly colocalized with the IMC protein GAP45 in the IMC (*Figure 1—figure supplement 2C*). The schizonts expressing Tb-IMC, Tb-cyto, or empty vector (EV: construct without ligase gene) were purified and treated with 100 μM biotin at 37°C for 3 hr. Both immunoblot and dot blot assays using streptavidin-HRP detected increased biotinylation in cell extracts in the presence of biotin from the Tb-IMC and Tb-cyto schizonts but not from the EV group (*Figure 1B* and *Figure 1—figure supplement 2D*). Furthermore, parasites stained with fluorescent-conjugated streptavidin (SA-488) and anti-HA antibody exhibited an IMC distribution (surrounding daughter merozoites) of biotinylated proteins, which colocalized with ligase in Tb-IMC schizonts. The biotinylated proteins and the ligase displayed cytosolic and nuclear distribution in the Tb-cyto schizonts, while scarce signal was detected in the EV schizonts (*Figure 1C* and *Figure 1—figure supplement 2E*). Therefore, the Tb-IMC enables the PL of IMC in the living schizonts.

To further confirm that the IMC or IMC-associated proteins were the primary targets of biotin labeling in the Tb-IMC schizonts, the protein extracts were subjected to streptavidin pull-down, followed by immunoblot assays. As expected, cis-biotinylation of the Tb-IMC ligase was detected and the IMC protein GAP45 was enriched in the pull-down fraction (*Figure 1—figure supplement 2F*). In contrast, the proteins of other organelles, including the MSP1 (PM), Erd2 (Golgi marker), BiP (ER marker), and histone H3 (nucleus) were not detected (*Figure 1—figure supplement 2F*). Three biological replicates were prepared from the Tb-IMC and Tb-cyto schizonts, and the streptavidin-affinity purified proteins from cell extracts were subjected to proteomic analyses by SWATH-MS, a data-independent acquisition based quantitative MS method (*Gillet et al., 2012*; *Li et al., 2015*). The numbers of identified proteins with at least two independent peptides were comparable between the Tb-IMC (replicate 1:1964 hits, replicate 2:1995 hits, and replicate 3:1986 hits) and Tb-cyto schizonts (replicate 1:1970 hits, replicate 2:1946 hits, and replicate 3:1948 hits). Correlation analyses of changes in protein abundance demonstrated good reproducibility among biological replicates (*Figure 1—figure supplement 2G*). Quantitative MS yielded 488 enriched proteins with high confidence (an adjusted p-value<0.05) in the Tb-IMC compared to the Tb-cyto schizonts (*Figure 1D* and *Figure 1—figure supplement 2H*).

Genes coding for IMC proteins display transcription peak at late schizont and merozoite during *P. falciparum* asexual replication cycle (*Hu et al., 2010*). To further filter potential IMC proteins, we discriminated the 488 Tb-IMC interacting proteins by comparative analyses of their transcription

pattern based on a *P. berghei* transcriptome dataset (*Otto et al., 2014*). After filtering out 188 proteins not showing transcription peak at schizont, we narrowed the candidates to 300 proteins (*Figure 1D*, *Figure 1—figure supplement 2H*, *Supplementary file 1*, and *Supplementary file 2*). These 300 Tb-IMC proximal interactors included many known IMC or IMC-associated proteins, including GAP40, GAP45, GAP50, MTIP, MyoA, ELC, GAPM1, GAPM2, GAPM3, IMCp, IMC1c, IMC1e, IMC1f, IMC1g, ISP3, BCP1, MORN1, CINCH, CDPK1, PhIL1, DHHC1, and DHHC2 (*Figure 1E and F*). The homologs of these proteins have displayed an IMC or IMC-like localization in the *Plasmodium*, or have been shown to interact or associate with IMC protein baits using immunoprecipitation assay in previous studies (*Bullen et al., 2009*; *Kumar et al., 2017*; *Hodson et al., 2015*; *Dearnley et al., 2012*; *Parkyn Schneider et al., 2017*; *Yeoman et al., 2011*; *Vaid et al., 2008*;). Therefore, the IMC and IMC-associated proteins were enriched in the list of 300 Tb-IMC interacting proteins, suggesting reliable data quality generated by TurboID and quantitative MS.

## Predicted functional profile of Tb-IMC interacting proteins

To further gain functional insights into the 300 Tb-IMC interacting proteins, we cross-referenced these proteins with other *Plasmodium* datasets. First, the Tb-IMC proximal interactors were significantly enriched for interactions using Search Tool for Retrieval of Interacting Genes/Proteins (STRING) (*Szklarczyk et al., 2019*), a database of known and predicted physical and functional protein-protein interactions. Notably, these proteins were segregated into two distinct subgroups (I and II) (*Figure 1E and F*). The subgroup I contained 152 proteins while the subgroup II contained 148 proteins. Most of known IMC or IMC-associated proteins were clustered into the subgroup I. In contrast, many annotated ER/Golgi secretory- or vesicle trafficking-related proteins were clustered into the subgroup II (*Figure 1E and F*). Second, gene ontology (GO) analysis was performed to further discriminate the proteins in subgroup I and II (*Figure 1—figure supplement 3*). Interestingly, the subgroup II was highly enriched with the GO terms, including the vesicle-mediated transport (40 proteins), vesicle fusion (11 proteins), ER to Golgi vesicle-mediated transport (13 proteins), Golgi vesicle transport (20 proteins) (*Figure 1—figure supplement 3*, biological process panel), and SNARE complex (14 proteins), endosome membrane (9 proteins), late endosome (7 proteins), and endocytic vesicle (16 proteins) (cellular component panel), while the subgroup I was mainly assigned with the GO terms, like the inner membrane pellicle complex (22 proteins) (cellular component panel), protein lipidation (10 proteins) (biological process panel), and palmitoyltransferase activity (6 proteins) (molecular function panel). The IMC arises de novo from ER/Golgi-derived material via vesicular trafficking and membrane fusion in the schizonts of each replication cycle of the parasite (*Gordon et al., 2008*; *Yeoman et al., 2011*; *Pieperhoff et al., 2013*; *Bannister et al., 2000*). Detection of the subgroup II enriched with vesicular trafficking and membrane fusion effectors agrees with the notion of the ER/Golgi-derived IMC biogenesis. In addition, subgroup I and II within the Tb-IMC interacting proteins may reflect the tight and dynamic association between IMC organelle and endomembrane system in schizont development.

## Validation of the candidate IMC proteins

To assess whether the identified proteins are indeed localized in the IMC, 21 candidates were selected among the 300 Tb-IMC interacting proteins for subcellular localization analysis (*Figure 2A*). These hits showed various levels at the enrichment ratio in the PL experiments. Among these proteins, the orthologues of 9 proteins including PY17X_ 0314700 (CDPK1), PY17X_0617900 (CDPK4), PY17X_1440500 (PKAr), PY17X_0839000 (PKAc), PY17X_1420600 (Rab11A), PY17X_1462100 (MTIP), PY17X_0206000 (PhIL1), PY17X_0525300 (GAPM2), and PY17X_1411000 (PIC5) have been experimentally validated to be IMC-residing or association in the schizonts of *P. berghei* or *P. falciparum* (*Patil et al., 2020*; *Bullen et al., 2009*; *Kumar et al., 2017*; *Saini et al., 2017*; *Parkyn Schneider et al., 2017*; *Fang et al., 2018*; *Green et al., 2006*; *Wilde et al., 2019*), while IMC localization or association of the 12 other candidates (PY17X_0207400, PY17X_0312400, PY17X_0417300, PY17X_0418000, PY17X_0917100, PY17X_1131200, PY17X_1139700, PY17X_1220300, PY17X_1348200, PY17X_1359500, PY17X_1441500, and PY17X_1453100) (*Figure 2B*) have not been well characterized in the *Plasmodium*. Among the 21 candidates, 6 out of them were endogenously tagged with a 6 HA at the C-terminus (*Figure 2—figure supplement 1A,B*), while others were tagged with a 6 HA at the N- or C-terminus and driven by the promoter of gene *isp3* for episomal expression in the asexual blood

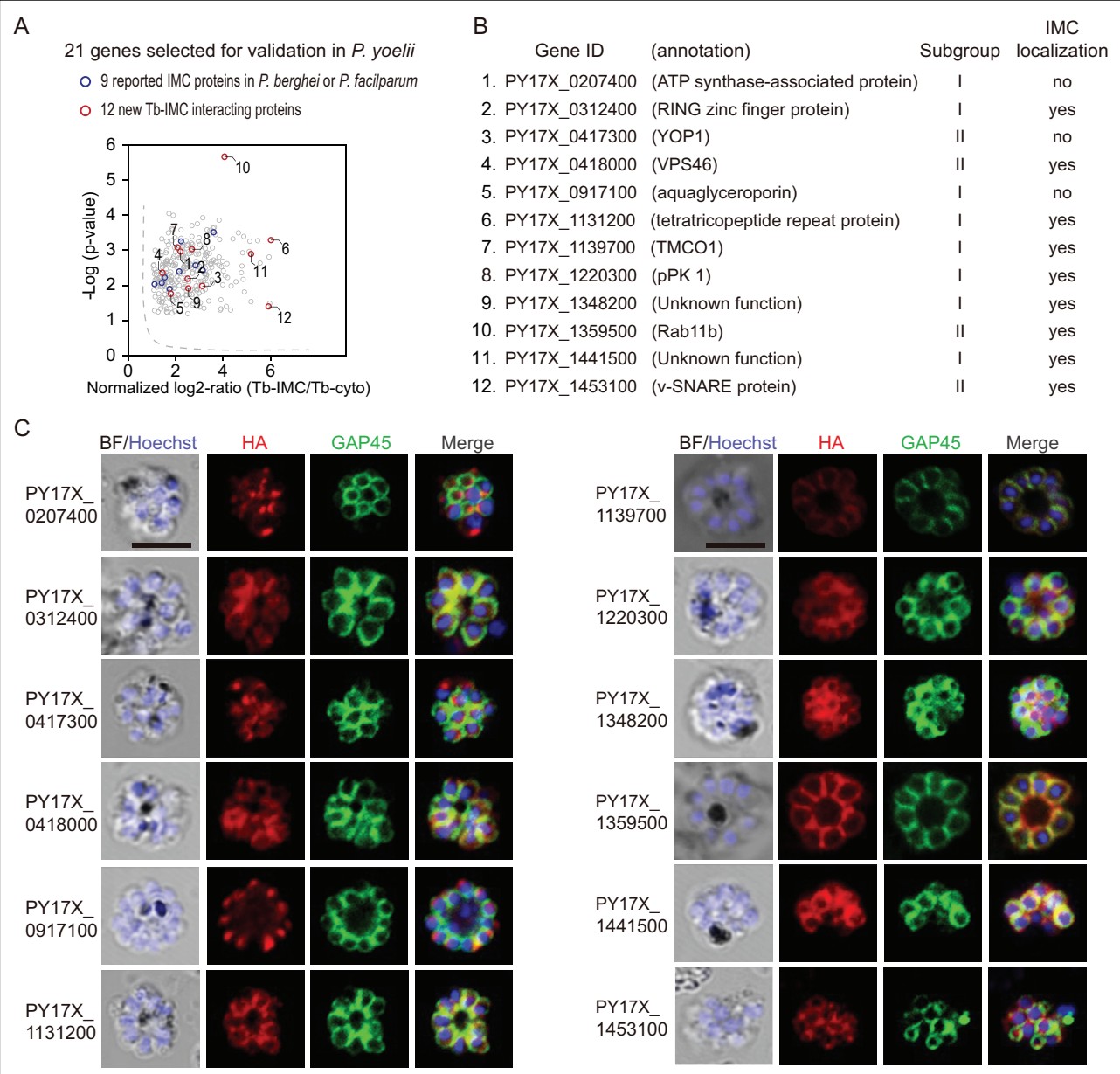

**Figure 2.** Validation of 9 new IMC proteins by localization analysis. (**A**) 21 candidates selected from the Tb-IMC interacting proteins for subcellular localization analysis in the *P. yoelii*. The orthologues of 9 proteins (blue dot) have been experimentally validated to be IMC-residing or -associated in the schizonts of *P. berghei* or *P. falciparum*, while subcellular localization of other 12 proteins (red dot) have not been well-characterized in *Plasmodium* species.(**B**) Information and subcellular localization summary of the 12 newly tested Tb-IMC interacting proteins shown in A. (**C**) Immunofluorescence assays (IFA) analysis of 12 Tb-IMC interacting proteins in the *P. yoelii* schizonts. Each protein was tagged with a 6 HA at the N- or C-terminus and episomally expressed in the schizonts. The schizonts were costained with antibodies against GAP45 and HA. Nuclei were stained with Hoechst 33342. Scale bar = 5 μm.

The online version of this article includes the following source data and figure supplement(s) for figure 2:

**Figure supplement 1.** Genotyping of genetically modified parasites.

**Figure supplement 1—source data 1.** (related to *Figure 2—figure supplement 1B*) Diagnostic PCR of Tb-IMC interacting protein candidates.

**Figure supplement 2.** Expression and localization analysis of IMC protein candidates selected in this study.

**Figure supplement 2—source data 1.** (related to *Figure 2—figure supplement 2A*) Immunoblot of 21 IMC protein candidates in the schizonts of *P. yoelii*.

**Figure supplement 3.** CDPK1 localizes at IMC in the schizonts of *P.yoelii*.

stages. Immunoblot assays were used to detect each protein, all displaying a band fitting their expected molecular weight (*Figure 2—figure supplement 2A*). As expected, immunofluorescence assays (IFA) showed clear colocalization of the 9 known proteins (CDPK1, CDPK4, PKAr, PKAc, Rab11A, MTIP, PhIL1, GAPM2, and PIC5) with the IMC marker GAP45 (*Figure 2—figure supplement 2B*). Additionally, we performed the ultrastructural expansion microscopy (U-ExM) and immune-EM to analyze the localization of CDPK1 in the *cdpk1::6 HA* schizonts and both methods detected the IMC localization of CDPK1 (*Figure 2—figure supplement 3A,B*). Among the 12 newly characterized candidates, 9 of them (PY17X_0312400, PY17X_0418000, PY17X_1131200, PY17X_1139700, PY17X_1220300, PY17X_1348200, PY17X_1359500, PY17X_1441500, and PY17X_1453100) displayed the IMC or IMC-like pellicle localization (*Figure 2C*), while 3 other candidates (PY17X_0207400, PY17X_0417300, and PY17X_0917100) did not. Collectively, we confirmed the IMC or IMC-like localization of 18 proteins from the 21 candidates in the *P. yoelii* schizonts.

## Palmitoylation of IMC proteins and regulation of localization

While a growing number of IMC proteins were discovered, their localization determinants remain incompletely known. Protein palmitoylation is a reversible lipid modification that facilitates protein attachment to the plasma and organelle membranes (*Fukata and Fukata, 2010*). Previous studies have shown that palmitoylation is important for the binding or targeting of proteins to IMC in the

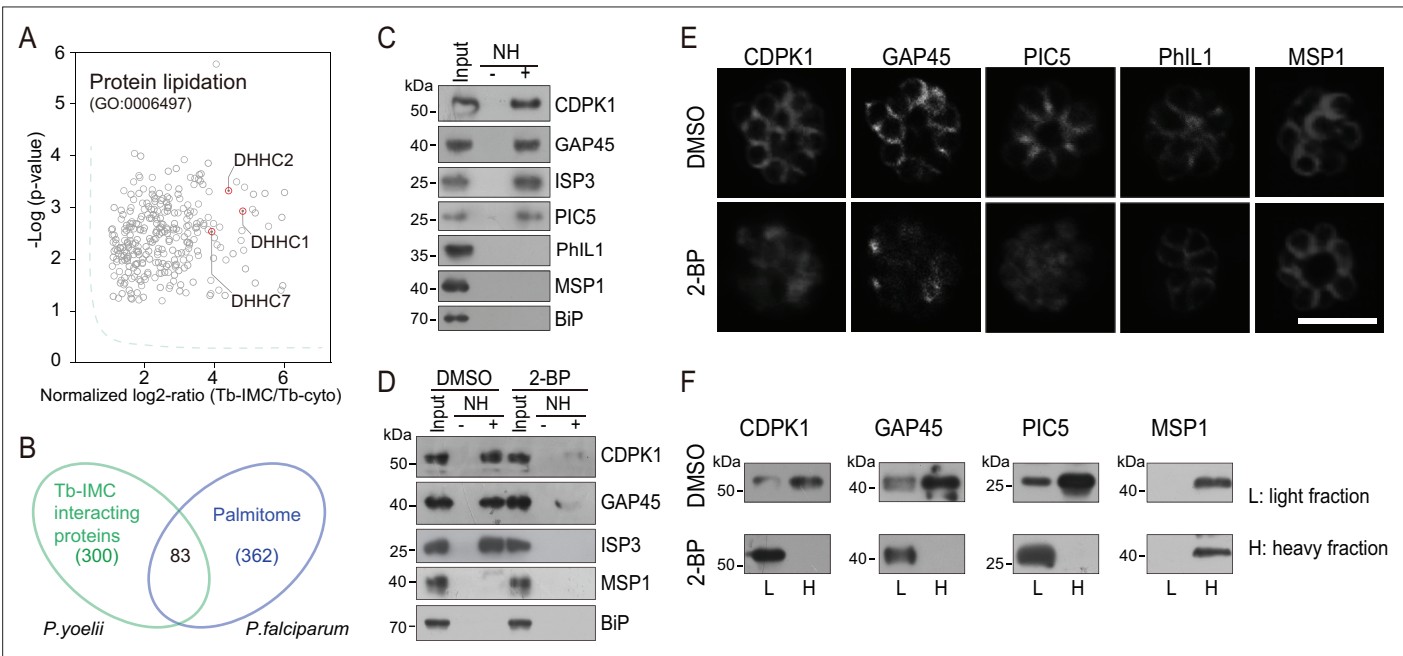

**Figure 3.** Palmitoylation regulates IMC protein localization. (**A**) Enrichment of enzymes for protein lipidation in the Tb-IMC interacting proteins shown in *Figure 1F*, including three palmitoyl-S-acyl-transferases DHHC2, DHHC1, and DHHC7. (**B**) Venn diagram showing overlap between Tb-IMC interactors (green) identified in this study and the orthologs within the *P. falciparum* palmitome (blue). 83 Tb-IMC interactors (overlap, 28%) were considered to be potentially palmitoylated. Numbers indicate the number of proteins identified. (**C**) Acyl-RAC method detecting palmitoylation of CDPK1, GAP45, ISP3, and PIC5, but not PhIL1 in schizonts. NH: NH$_2$OH. MSP1 and BiP served as loading controls. Total proteins were treated with MMTs to block the thiol side chain in free cysteine. Proteins with palmitoylated cysteine were re-exposed the thiol side chain with removal of palmitic acid by NH$_2$OH, purified via Thiopropyl Sepharose, and eluted by DTT for immunoblot. Representative of two independent replicates. (**D**) Palmitoylation analysis of CDPK1, GAP45, and ISP3 in schizonts treated with 2-BP. NH: NH$_2$OH. MSP1 and BiP served as loading controls. Representative of two independent replicates. (**E**) Immunofluorescence assays (IFA) analysis of CDPK1, GAP45, PIC5, PhIL1, and MSP1 in schizonts treated with 2-BP and DMSO, respectively. Purified mature schizonts were treated with 2-BP for 12 hr. Scale bar = 5 μm. (**F**) Fractionation analysis of CDPK1, GAP45, PIC5, and MSP1 in schizonts treated with 2-BP and DMSO, respectively. Light fraction includes cytosolic proteins while heavy fraction includes membrane proteins and cytoskeleton proteins. Representative of two independent replicates.

The online version of this article includes the following source data for figure 3:

**Source data 1.** (related to *Figure 3C*) Acyl-RAC method detecting palmitoylation of CDPK1, GAP45, ISP3, and PIC5, but not PhIL1 in schizonts.

**Source data 2.** (related to *Figure 3D*) Palmitoylation analysis of CDPK1, GAP45, and ISP3 in schizonts treated with 2-BP.

**Source data 3.** (related to *Figure 3F*) Fractionation analysis of CDPK1, GAP45, PIC5, and MSP1 in schizonts treated with 2-BP and DMSO, respectively.

*Plasmodium* (*Wang et al., 2020*; *Wetzel et al., 2015*). We speculated that certain IMC proteins use lipid moieties for IMC membrane attachment. Interestingly, we observed a significant enrichment of the biological-process term of protein lipidation (GO: 0006497) in the Tb-IMC interacting proteins (*Figure 3A* and *Figure 1—figure supplement 3*). Three palmitoyl-S-acyl-transferases (PAT) including DHHC1, DHHC2, and DHHC7 were enriched (*Figure 3A*), implying a role of these PATs for palmitoylation of the IMC proteins.

To identify IMC proteins with palmitoylation, we examined a collection of 494 palmitoylated proteins previously detected in the schizonts of *P. falciparum* (*Jones et al., 2012*). Among the 300 *P. yoelii* Tb-IMC interacting proteins identified in this study, 83 proteins (28%) have orthologs that are palmitoylated in *P. falciparum* (*Figure 3B* and *Supplementary file 1*). Importantly, these 83 proteins includes GAP45 and ISP3, whose palmitoylations have been experimentally validated (*Wang et al., 2020*; *Jones et al., 2012*), and CDPK1, GAP50, IMC1g, and IMC1c, which are predicted to be palmitoylated. Out of these 83 proteins, we assessed 4 (CDPK1, GAP45, ISP3, and a newly identified IMC protein PIC5) for their palmitoylation using the resin-assisted capture of acylated proteins (Acyl-RAC) method (*Forrester et al., 2011*). Palmitoylation was confirmed in these 4 proteins in *P. yoelii* schizonts (*Figure 3C*), while PhIL1, an IMC protein not in the list of 83 proteins, did not display palmitoylation (*Figure 3C*). To further confirm palmitoylation, mature schizonts was treated with 100 μM 2-bromopalmitate (2-BP), an inhibitor of protein palmitoylation. We observed markedly reduced palmitoylation of CDPK1, GAP45, and ISP3 in the 2-BP treated schizonts (*Figure 3D*). Notably, CDPK1, GAP45, and PIC5 lost their IMC localization and were found in the cytosol or nucleus after 2-BP treatment (*Figure 3E*). Lower fluorescence signal of GAP45 and CDPK1 after 2-BP treatment may be due to loss of protein concentration at the IMC. As control, the IMC localization of PhIL1 and the PM localization of merozoite surface protein MSP1 were unaffected. Fractionation of schizont protein extracts (*Figure 3F*) revealed that CDPK1, GAP45, and PIC5 were mainly present in the heavy fraction, in agreement with the IMC membrane association of these proteins. However, these proteins were mostly detected in the light fraction after 2-BP treatment, suggesting their distribution alterations after losing the palmitoylation. Together, these results suggest that palmitoylation may exist in a relatively high proportion of the IMC proteins and that it is important for subcellular localization of certain IMC proteins including CDPK1, GAP45, and PIC5.

## DHHC2 is an IMC-residing palmitoyl-S-acyl-transferase in schizonts

Next, we searched for the PATs that catalyze the palmitoylation of IMC residing proteins in the schizonts. 11 putative PATs (named DHHC1–11) were predicted in the genomes of rodent malaria parasites (*Hodson et al., 2015*). Quantitative reverse transcription-PCR (qRT-PCR) analysis revealed that the *P. yoelii dhhc2* displayed the highest mRNA level in the schizonts (*Figure 4A*). The mRNA levels of these *P. yoelii dhhc* genes were positively correlated ($R^2$=0.94) with the transcription profiles of their orthologs in *P. berghei* determined via RNA-seq (*Otto et al., 2014*; *Figure 4A*). Furthermore, we analyzed the localization of the 11 *P. yoelii* endogenous PATs individually in the schizonts of transgenic strains previously generated (*Wang et al., 2020*). Out of the 11 PATs, only DHHC1 and DHHC2 displayed clear IMC localization in the schizonts with a stronger IFA signal for DHHC2 (*Figure 4B* and *Figure 4—figure supplement 1A*). Immunoblot analysis of protein extracts from the same number of schizonts of the endogenously tagged *dhhc2::6* HA and *dhhc1::6* HA parasites revealed approximately fivefold higher level of DHHC2 than DHHC1 (*Figure 4C*). Interestingly, DHHC2 but not DHHC1 displayed an interrelation with several IMC proteins, including CDPK1 and GAP45, in a STRING analysis which predicts the likelihood of protein-protein interaction (*Figure 4—figure supplement 1B*). The IMC localization of DHHC2 was confirmed in two independent strains *6 HA::dhhc2* and *dhhc2::4Myc*, whose endogenous DHHC2 was tagged with an N-terminal 6 HA and C-terminal 4Myc tag, respectively (*Figure 4D*). Immunoblot analyses of the membrane and cytoplasmic fractions of schizont lysates also revealed that DHHC2 was mainly detected in the membrane fraction (*Figure 4E*). As a control, merozoite PM protein MSP1 was detected in the membrane fraction while the cytosolic protein GAPDH was in the cytoplasmic fraction (*Figure 4E*).

To investigate the expression dynamics of DHHC2 during parasite development, early and later stages of intraerythrocytic *dhhc2::6* HA parasites were isolated using the Nycodenz gradient centrifugation. Immunoblot showed that DHHC2 is highly expressed in the late trophozoites and schizonts, but not in the rings or early trophozoites (*Figure 4F*). IFA also revealed DHHC2 signal in the early and

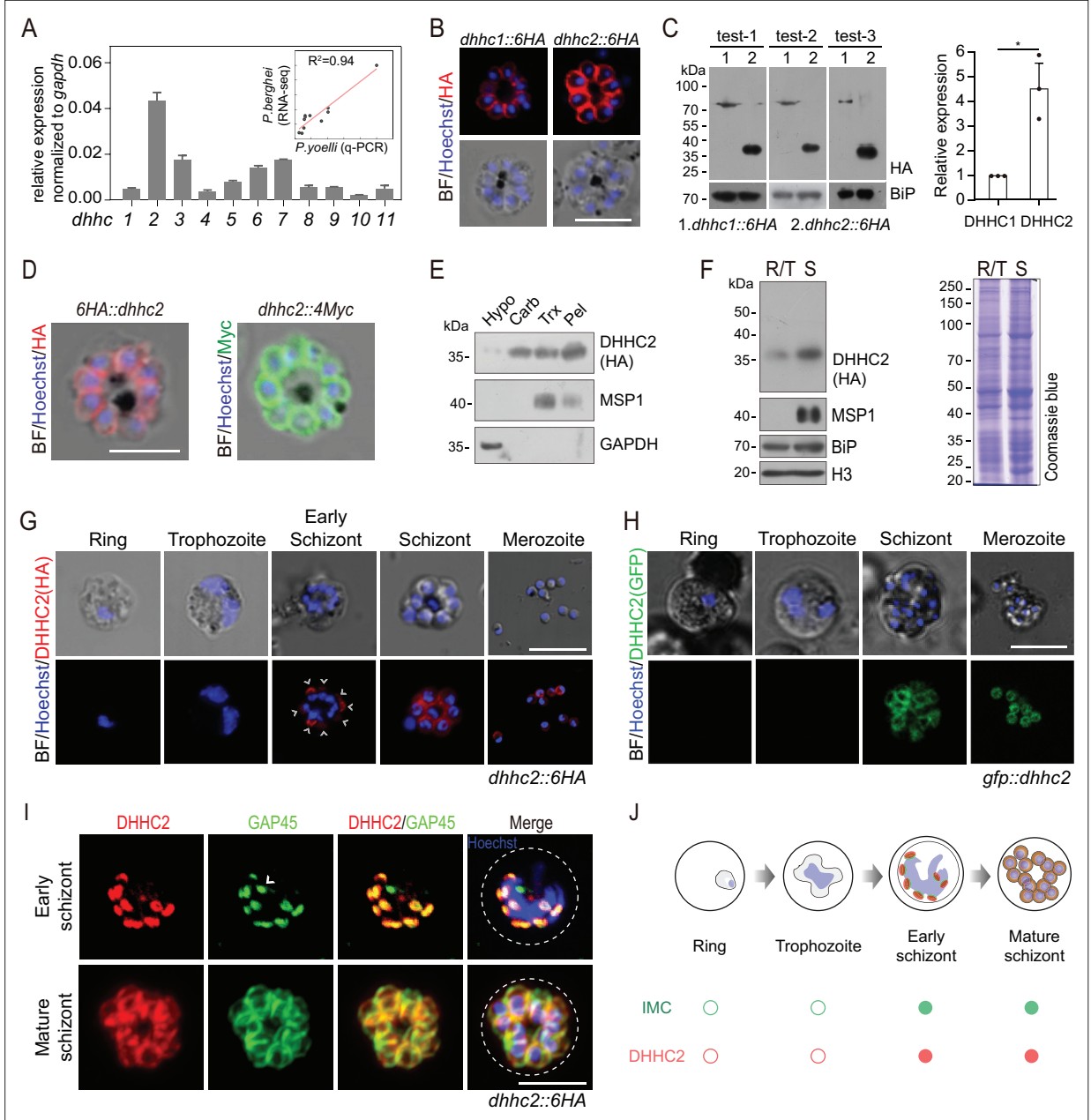

**Figure 4.** DHHC2 is an IMC-residing palmitoyl-S-acyl-transferase in schizonts. (**A**) RT-qPCR of transcripts for 11 *dhhc* (*dhhc1- dhhc11*) in schizonts. Gene expression was normalized to the *gapdh* transcript. The inlet indicates positive correlation between mRNA levels of these *P. yoelii dhhc* genes and mRNA levels of their *dhhc* orthologs in *P. berghei* determined via RNA-seq. Values are means ± SD (n=3). (**B**) Immunofluorescence assays (IFA) analysis of DHHC1 and DHHC2 in schizonts from two tagged parasite strains *dhhc1::6* HA and *dhhc2::6* HA. Scale bar = 5 μm. (**C**) Immunoblot of DHHC1 and DHHC2 from the cell lysate of similar number of schizonts from the *dhhc1::6* HA and *dhhc2::6* HA parasite, respectively. Right panel: the quantification of band intensity. Values are means ± SEM (n=3 biological replicates), two-tailed *t*-test, *p<0.05. (**D**) IFA analysis of DHHC2 in schizonts of another two tagged parasite strains *6 HA::dhhc2* and *dhhc2::4Myc*. Scale bar = 5 μm. (**E**) Solubility assay detected membrane association of DHHC2 in schizonts using different detergents. Cytosolic soluble proteins are in hypotonic buffer (Hypo), peripheral membrane proteins in carbonate buffer (Carb), integral membrane proteins in Triton X-100 buffer (Trx), and insoluble proteins in pellet (Pel). DHHC2 is in the membrane-associated fractions as IMC protein GAP45 and PM protein MSP1, while cytoplasm protein GAPDH is in the soluble fraction. Representative of two independent replicates. (**F**) Immunoblot of DHHC2 from early stages containing ring and trophozoite (R/T) and late stages containing schizont (S) of the *dhhc2::6* HA parasites. Merozoite surface protein MSP1 was mainly expressed in the schizonts. BiP, histone H3, and Coomassie blue staining of total lysate were used as loading control. Representative of two independent replicates. (**G**) DHHC2 expression dynamics in different asexual blood stages (ring, trophozoite, schizont, and merozoite) of the parasite *dhhc2::6* HA by IFA. Scale bar = 5 μm. (**H**) Fluorescent microscopy of GFP::DHHC2 in different asexual blood stages of the parasites *gfp::dhhc2*. Scale bar = 5 μm. (**I**) Maximum intensity projections of super-resolution immunofluorescence microscopy (Airyscan) of early and

*Figure 4 continued on next page*

Figure 4 continued

mature *dhhc2::6* HA schizonts stained with anti-HA and anti-GAP45 antibodies. Arrow in the graph indicated a 'GAP45-only spot', which likely resulted from the difference in protein stain efficiency for DHHC2 and GAP45. Scale bar = 5 µm. (**J**) Model showing the IMC-associated localization of DHHC2 in the schizonts.

The online version of this article includes the following source data and figure supplement(s) for figure 4:

**Source data 1.** (related to *Figure 4A*) RT-qPCR of transcripts for 11 *dhhc* (dhhc1- dhhc11) in schizonts.

**Source data 2.** (related to *Figure 4C*) Immunoblot of DHHC1 and DHHC2 from the cell lysate of similar number of schizonts from the dhhc1::6 HA and dhhc2::6 HA parasite, respectively.

**Source data 3.** (related to *Figure 4C*) Quantification of DHHC1 and DHHC2 intensity in immunoblot.

**Source data 4.** (related to *Figure 4E*) Solubility assay detected membrane association of DHHC2 in schizonts using different detergents.

**Source data 5.** (related to *Figure 4F*) Immunoblot of DHHC2 from early stages containing ring and trophozoite (R/T) and late stages containing schizont (**S**) of the dhhc2::6 HA parasites.

**Figure supplement 1.** Localization analysis of 11 PATs (DHHC1-11) in schizonts.

**Figure supplement 1—source data 1.** (related to *Figure 4—figure supplement 1D*) Diagnostic PCR of gfp::dhhc2 parasite line.

mature schizonts and free merozoites (*Figure 4G*). These observations were independently confirmed in another strain *gfp::dhhc2*, in which endogenous DHHC2 was tagged with an N-terminal GFP (*Figure 4H* and *Figure 4—figure supplement 1C,D*). The expression dynamics of DHHC2 is consistent with the IMC biogenesis in the schizonts. Using the Airyscan microscopy, we observed clear colocalization between DHHC2 and GAP45 in both early and mature schizonts (*Figure 4I*). Interestingly, both proteins were found as separate dots that colocalize at the periphery of the intact nucleus in the early schizonts (*Figure 4I*), suggesting that IMC arises de novo at multiple points at or close to the nucleus. Together, these results indicate that DHHC2, as an IMC-residing PAT, is expressed throughout the IMC biogenesis in the schizonts of *P. yoelii* (*Figure 4J*).

## DHHC2 is essential for the asexual blood stage development in mice

DHHC2 has been suggested to play an essential role in the asexual blood stage development of the parasites since no viable mutant clone was obtained using either the conventional or Cas9-based knockout strategies in the *P. yoelii* and *P. berghei* (*Wang et al., 2020*; *Santos et al., 2015*). To explore the functions of DHHC2, we applied an Auxin-inducible degron (AID)-based protein degradation system in the *P. yoelii* transgenic strain *Tir1* (*Liu et al., 2020*), which allows depletion of the target protein fused to a miniAID (mAID) motif with the aid of the plant hormone auxin (Indole-3-acetic acid, IAA). The N-terminus of the endogenous *dhhc2* locus was tagged with the sequence encoding mAID::2 HA in the *Tir1* strain, generating the *mAID::dhhc2* clone (*Figure 5—figure supplement 1A*). This parasite displayed normal proliferation during asexual blood stages and the fusion protein mAID::DHHC2 exhibited IMC localization in the schizonts (*Figure 5—figure supplement 1B*), indicating no detrimental effect of mAID tagging on DHHC2 localization and function. IAA treatment (1 mM for 3 hr) of the *mAID::dhhc2* schizonts efficiently depleted the mAID::DHHC2 protein (*Figure 5—figure supplement 1C*). To determine whether IAA itself affects parasites development in vivo, mice infected with the 17XNL parasite were injected intraperitoneally with 200 mg/kg/day IAA or vehicle (DMSO) for 3 consecutive days. The in vivo parasitemia increased at an indistinguishable rate in both groups (*Figure 5—figure supplement 1D*), indicating no notable effect of IAA on parasite proliferation in mice. Next, we tested whether the parasite mAID::DHHC2 protein could be depleted in mice. The mice with ~10% parasitemia of the *mAID::dhhc2* parasite were injected intraperitoneally with IAA once and the parasite-infected red blood cells were collected for immunoblot at 1 and 3 hr after IAA injection (*Figure 5—figure supplement 1E*). The mAID::DHHC2 protein was significantly reduced in the parasites from IAA-treated mice, indicating successful mAID::DHHC2 degradation by IAA (*Figure 5—figure supplement 1E*). As a control, the IAA treatment had little effect on the *6 HA::DHHC2* protein in the *6 HA::dhhc2* parasite (*Figure 5—figure supplement 1E*).

To dissect the DHHC2 function in vivo, mice were infected with the *Tir1* or *mAID::dhhc2* schizonts which were pretreated with IAA or vehicle for 3 hr in vitro to deplete DHHC2 (*Figure 5A*). From 12 hr postinfection, the parasitemia in mice infected with *Tir1* and *mAID::dhhc2* was monitored in parallel every 12 hr. The parasitemia of *Tir1* increased at an equal rate after either IAA or vehicle pretreatment (*Figure 5B*, left panel). However, the IAA-pretreated *mAID::dhhc2* parasite displayed delayed

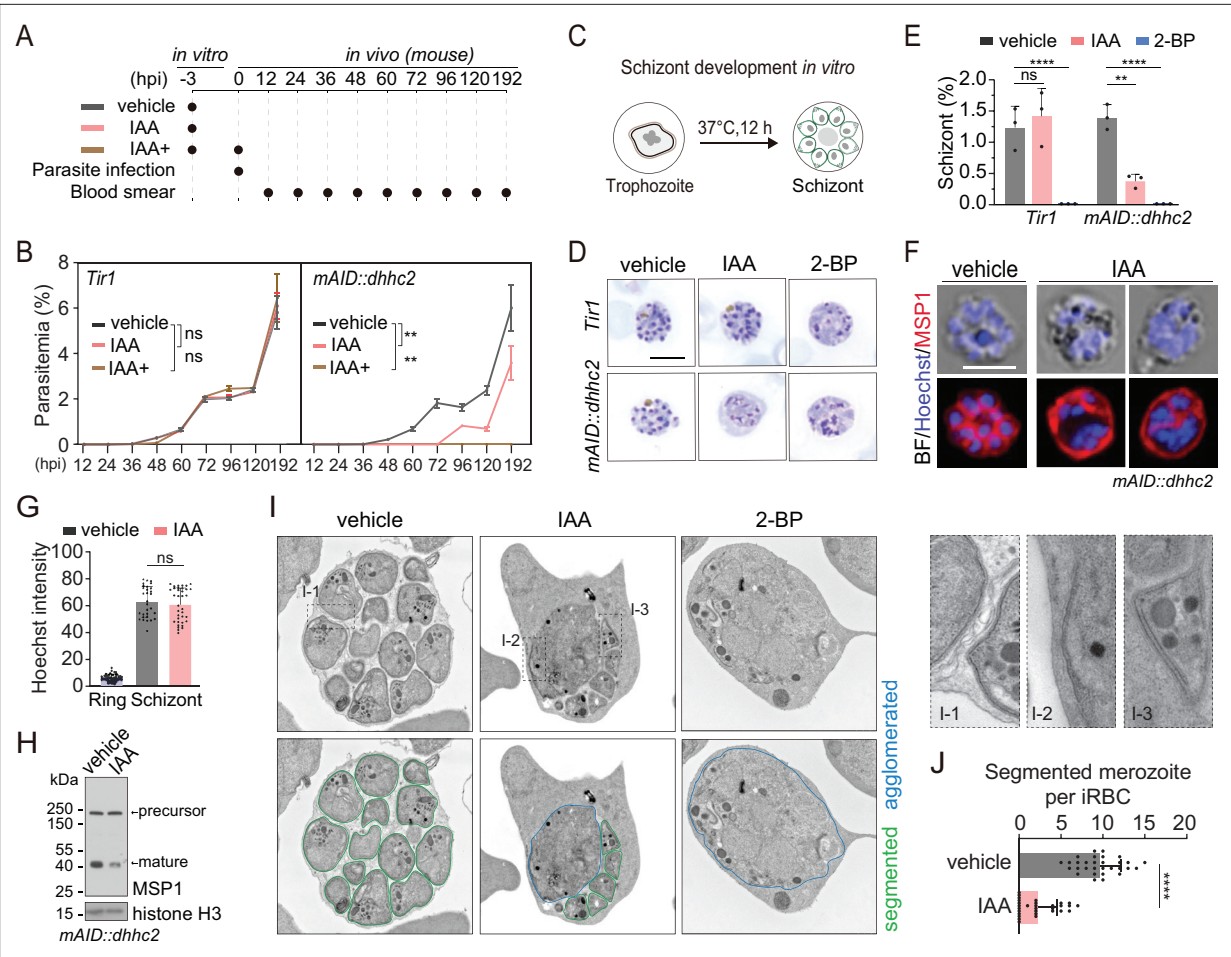

**Figure 5.** DHHC2 is essential for parasite proliferation in mice and regulates schizont segmentation and merozoite invasion. (**A**) Experimental design of in vivo test of DHHC2 essentiality using AID. Parasites of *Tir1* and *mAID::dhhc2* were pretreated with vehicle or indole-3-acetic acid (IAA) for 3 hr in vitro, then intravenously injected to C57BL/6 mice. In the IAA+ treatment group, another IAA injection (200 mg/kg, *ip*) at time of parasite infection was applied for further DHHC2 depletion in vivo. From 12–192 hr postinfection, the parasitemia in mice infected with *Tir1* and *mAID::dhhc2* was monitored by blood smear every 12 hr. (**B**) *Tir1* and *mAID::dhhc2* parasite proliferation in mice (n=3 per group) at each treatment group in (**A**), two-tailed *t*-test, **p<0.01, ns, not significant. (**C**) Schematic of the schizont development from trophozoite in vitro. Purified early stage parasites including ring and trophozoite were cultured with vehicle, IAA, or 2-BP for 12 hr for schizont development. (**D**) Giemsa staining of the schizonts developed from *Tir1* and *mAID::dhhc2* parasites treated with vehicle, IAA, or 2-BP illustrated in (**C**). Scale bar = 5 µm. (**E**) Quantification of mature schizonts in (**D**). Purified early stage parasites including ring and trophozoite were treated with IAA or 2-BP for 12 hr. Values are means ± SEM (n=3 biological replicates), two-tailed *t*-test, **p<0.01, ***p<0.001. (**F**) Costaining of the *mAID::dhhc2* schizonts in (**C**) with antibody against merozoite surface protein MSP1 and Hoechst 33342. Scale bar = 5 µm. (**G**) Quantification of Hoechst signal in schizonts indicating nuclear DNA contents in (**F**). More than 30 schizonts were analyzed in each group. Signal in ring stage parasites serves as a control. Values were means ± SD, Mann–Whitney U test applied, ns, not significant. (**H**) Immunoblot of MSP1 in the schizonts developed from the *mAID::dhhc2* parasites treated with vehicle and IAA. Precursor form (~200 kD) and mature form (~42 kD) of MSP1 were shown. Histone H3 used as a loading control. (**I**) Representative images of transmission electron microscopy (TEM) of schizonts developed from the *mAID::dhhc2* parasites treated with vehicle, IAA, or 2-BP. Purified early stage parasites including ring and trophozoite were treated with IAA or 2-BP for 12 hr. Right panels indicate three examples of representative daughter cell pellicle including plasma membrane (PM) and inner membrane complex (IMC). (**J**) Quantification of fully segmented merozoites in schizonts in (**I**). More than 30 schizonts were analyzed in each group. Values were shown as means ± SD, Mann–Whitney U test, ****p<0.0001.

The online version of this article includes the following source data and figure supplement(s) for figure 5:

**Source data 1.** (related to *Figure 5B*) Tir1 and mAID-dhhc2 parasite proliferation in mice.

**Source data 2.** (related to *Figure 5E*) Quantification of mature schizonts.

**Source data 3.** (related to *Figure 5G*) Quantification of Hoechst signal in schizonts.

**Source data 4.** (related to *Figure 5H*) Immunoblot of MSP1 in the schizonts developed from the mAID::dhhc2 parasites treated with vehicle and IAA.

**Source data 5.** (related to *Figure 5J*) Quantification of fully segmented merozoites in schizonts.

*Figure 5 continued on next page*

*Figure 5 continued*

**Figure supplement 1.** Generation of the modified strains with endogenous DHHC2 and DHHC2 tagged with a mAID motif for induced protein degradation.

**Figure supplement 1—source data 1.** (related to *Figure 5—figure supplement 1A*) Diagnostic PCR of *dhhc1::maid, mid::dhhc2* parasite lines.

**Figure supplement 1—source data 2.** (related to *Figure 5—figure supplement 1C*) Immunoblot of fusion proteins DHHC1::mAID (left panel) and mAID::DHHC2 (right panel) in the schizonts of the Tir1, dhhc1::mAID and mAID::dhhc2 parasites treated with vehicle or IAA (1 mM) for 3 hr.

**Figure supplement 1—source data 3.** (related to *Figure 5—figure supplement 1D*) Proliferation assessment of wildtype parasite in mice treated with IAA.

**Figure supplement 1—source data 4.** (related to *Figure 5—figure supplement 1E*) IAA-induced degradation assessment of parasite mAID::DHHC2 protein in mice.

**Figure supplement 2.** DHHC2 depletion impaired merozoite invasion.

**Figure supplement 2—source data 1.** (related to *Figure 5—figure supplement 2B*) The invasion efficiency of merozoite in the mice evaluated by blood smear.

**Figure supplement 2—source data 2.** (related to *Figure 5—figure supplement 2C*) The invasion efficiency of merozoite in the mice evaluated by flow cytometry.

proliferation compared to the parasite pretreated with vehicle (*Figure 5B*, right panel). The parasite with IAA-pretreatment emerged in the mouse blood at 96 hr postinfection while the parasite with vehicle-pretreatment emerged at 36 hr. Notably, continuation of DHHC2 depletion by another IAA injection (IAA+) at time of parasite infection resulted in complete suppression of *mAID::dhhc2* in mice (*Figure 5B*, right panel), while this treatment had no effect on the proliferation of *Tir1* (*Figure 5B*, left panel). These results provided a direct evidence that DHHC2 is essential for the asexual blood stage development in mice.

## DHHC2 regulates schizont segmentation

Because DHHC2 is specifically expressed in the schizonts, we speculated that DHHC2 regulates schizont development. A mixture of early stage parasites (rings and early trophozoites) was purified using Nycodenz centrifugation and cultured for 12 hr to mature schizonts using an in vitro culture method (*Janse et al., 2006*; *Figure 5C*). We evaluated the effect of DHHC2 depletion on schizont development by counting the mature schizonts using Giemsa staining. The IAA-treated *Tir1* parasite developed to mature schizonts at a similar level as the vehicle-treated *Tir1* parasite, indicating no effect of IAA alone on the schizont development (*Figure 5D and E*). However, IAA treatment severely decreased the formation of mature schizonts in the *mAID::dhhc2* parasite (*Figure 5D and E*). Importantly, the treated *mAID::dhhc2* parasite had apparent nuclear replication but failed to segregate into individual merozoites. This result suggests arrest of schizont cytokinesis or segmentation after nuclear multiplication. In addition, treatment with the PAT inhibitor 2-BP resulted in no formation of mature schizonts in either *Tir1* or *mAID::dhhc2* parasites (*Figure 5D and E*), which is in agreement with the previous reports (*Jones et al., 2012*).

To further examine the defects within the schizonts, the *mAID::dhhc2* parasite was costained with the nuclear dye and an antibody against the merozoite surface protein MSP1, which coats the parasite PM of newly formed daughter merozoites after segmentation (*Blackman et al., 1994*). Hoechst signals in the schizonts showed no notable difference between the IAA- and vehicle-treated groups (*Figure 5F and G*), indicating normal nuclear DNA multiplication in the DHHC2-deficient (IAA-treated) parasites. However, MSP1 staining revealed that the schizonts of DHHC2-deficient group had agglomerates of daughter cells that failed to separate (*Figure 5F*) while the schizonts of the vehicle control group exhibited normal morphology (*Figure 5F*). Expression and cleavage (from ~200 to 42 kD) of full length MSP1 have been used to indicate schizont maturation and merozoite ready-to-egress from erythrocytes (*Silmon de Monerri et al., 2011*; *Child et al., 2010*). Consistent with an impaired schizont development, an immunoblot of the IAA-treated parasite showed reduced MSP1 expression and processing compared to the vehicle control (*Figure 5H*).

Next, using transmission electron microscopy (TEM), we examined the ultrastructure of the *mAID::dhhc2* schizonts produced after the 12 hr in vitro maturation in the presence of the vehicle, IAA, or 2-BP. The daughter cells were normally segmented in the schizonts of the vehicle-treated group, forming fully separated daughter merozoites (*Figure 5I*). However, severe morphological

defects occurred in the DHHC2-deficient parasites. Large daughter cell agglomerates were observed in the center of most schizonts although a few mononucleated daughter merozoites were formed (*Figure 5I*). This segmentation arrest resulted in significant fewer daughter merozoites in the DHHC2-deficient schizonts compared to the control (*Figure 5J*). Interestingly, we observed the incomplete IMC beneath the PM in some daughter merozoites and the agglomerates within the DHHC2-deficient schizonts (*Figure 5I*), suggesting no remarkable effect of DHHC2 depletion in the initial biogenesis of the IMC. In addition, the organelles including possible rhoptries or micronemes were observed in the large agglomerates of the DHHC2-deficient schizonts, suggesting normal biogenesis and development of these organelles but a defect in allocation of daughter merozoites during schizont segmentation. As a control, the 2BP treatment completely blocked schizont segmentation (*Figure 5I*).

## DHHC2 also controls merozoite invasion

Upon schizont maturation, merozoites egress from the erythrocyte to invade new erythrocytes. To investigate DHHC2 function in merozoite invasion, we collected the merozoites released from the mechanically disrupted mature schizonts, which undergo natural rupture in an extremely low efficacy under the in vitro condition for the *P. yoelii*. Released merozoites were capable of invading the erythrocytes in mice, indicative of merozoite's viability and activity. The merozoites collected from the IAA- or vehicle-treated schizonts were injected intravenously into mice and the number of the ring stage parasites indicative of successful invasion was counted by flow cytometry and light microscopy (*Figure 5—figure supplement 2A*). At 20 min postinjection, the number of newly developed rings was significantly lower in the IAA-treated group compared with the vehicle-treated group (*Figure 5—figure supplement 2B,C*). As a control, the IAA treatment had no effect on merozoite invasion of the *Tir1* parasite (*Figure 5—figure supplement 2B,C*). Taking all the results together, DHHC2 has an important function in both schizont segmentation and merozoite invasion, two processes during which the parasites possess IMC.

## DHHC2 palmitoylates GAP45 and CDPK1

GAP45 and CDPK1 are essential for the asexual blood stage development of *P. falciparum* (*Perrin et al., 2018*; *Kumar et al., 2017*; *Azevedo et al., 2013*). Additionally, palmitoylation regulates IMC targeting of these proteins in the schizonts and merozoites (*Figure 3E*). We speculated that DHHC2 exerts its function via palmitoylating GAP45 and CDPK1. To test this hypothesis, the palmitoylation and localization of GAP45 and CDPK1 were examined in the DHHC2-deficient schizonts after pretreating with IAA at 1 mM for 12 hr. The palmitoylation level of GAP45 and CDPK1 was significantly reduced in the DHHC2-deficient schizonts compared to the vehicle control (*Figure 6A*). Notably, both GAP45 and CDPK1 lost their typical IMC localization in both the schizonts and the released merozoites of the DHHC2-deficient parasite (*Figure 6C and D*). In contrast, PhIL1 retained the IMC localization in the DHHC2-deficient parasite (*Figure 6C* and *Figure 6—figure supplement 1A,B*), consistent with the fact that PhIL1 is not palmitoylated (*Figure 3C*). The distribution of MSP1 in the DHHC2-deficient schizonts was also unaffected (*Figure 6C*). In agreement with the changes in subcellular distribution of proteins by IFA, the detergent extraction-based protein solubility assay also revealed that GAP45 and CDPK1 lost their membrane association upon the depletion of DHHC2 by IAA (*Figure 6E*).

Besides DHHC2, DHHC1 also displayed IMC localization in the schizonts although it is expressed at lower abundance (*Figure 4B and C*). To investigate whether DHHC1 also contributes to the palmitoylation of GAP45 and CDPK1, we generated the *dhhc1::mAID* parasite clone in which the C-terminus of endogenous DHHC1 was tagged with the mAID::HA module in the *Tir1* strain (*Figure 6—figure supplement 1A,B*). IAA treatment depleted the DHHC1::mAID protein in the *dhhc1::mAID* schizonts (*Figure 6—figure supplement 1C*), but had little impact on the palmitoylation level of GAP45 and CDPK1 (*Figure 6B*). These results indicated that DHHC2, but not DHHC1, contributes to the palmitoylation of GAP45 and CDPK1 in the schizonts.

To further confirm DHHC2 as the enzyme capable of palmitoylating GAP45 and CDPK1, we transfected constructs encoding HA-tagged human codon optimized DHHC2 (DHHC2-HA), GAP45 and CDPK1 into the human HEK293T cells. Indeed, the ectopically expressed GAP45 and CDPK1 could be palmitoylated by cotransfected DHHC2-HA, while the PAT catalytic-deficient mutant protein DHHC2/C128A-HA failed to palmitoylate GAP45 and CDPK1 (*Figure 6F*). As a control, the palmitoylation of cotransfected mouse CD36 protein, which was reported to be constitutively palmitoylated in HEK293T

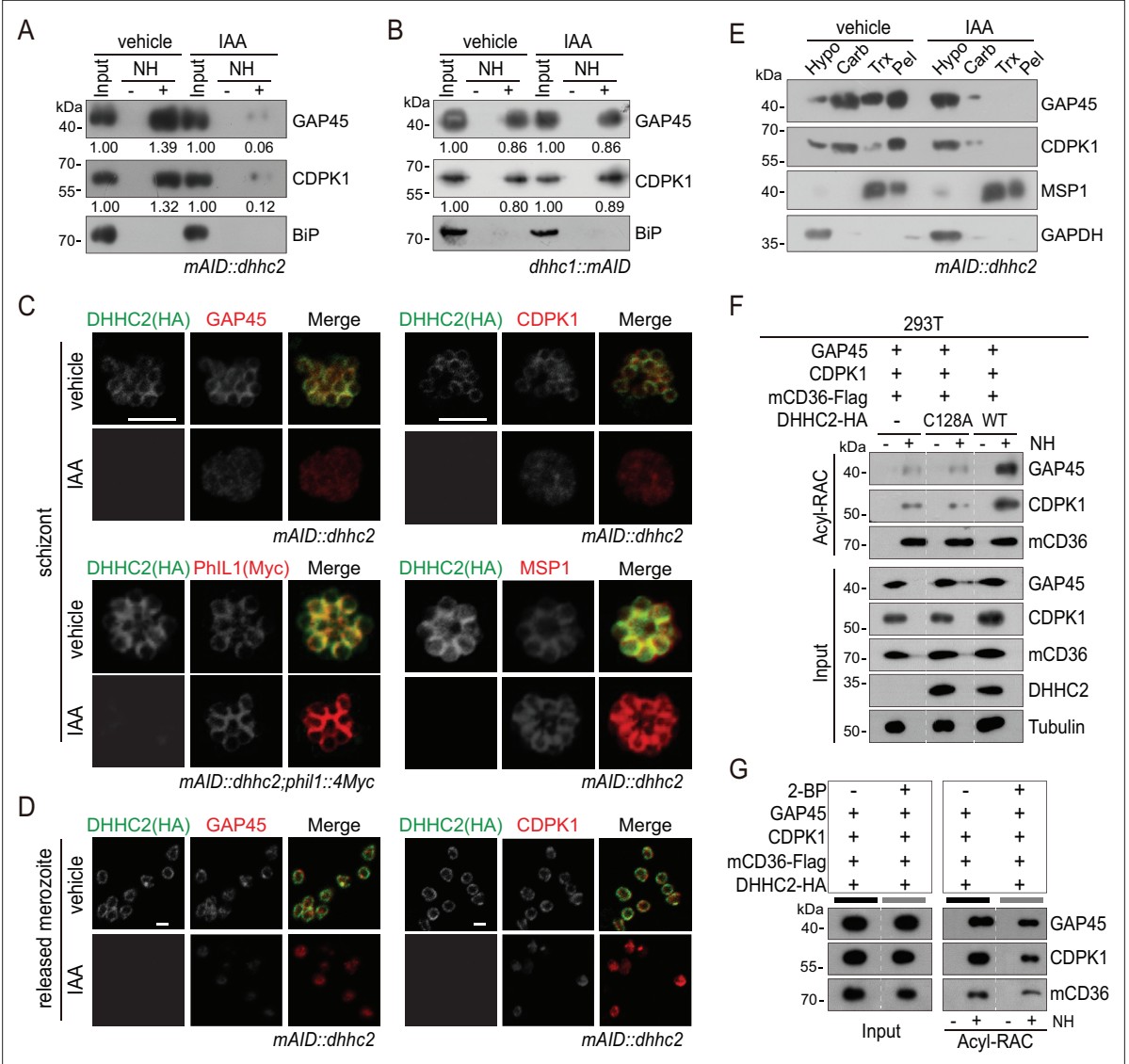

**Figure 6.** DHHC2 palmitoylates GAP45 and CDPK1 in the schizonts. (**A**) Acyl-RAC method detecting palmitoylation of GAP45 and CDPK1 in the *mAID::dhhc2* schizonts treated with vehicle or indole-3-acetic acid (IAA). BiP served as a loading control. Representative of two independent replicates. (**B**) Acyl-RAC method detecting palmitoylation of GAP45 and CDPK1 in the *dhhc1::mAID* schizonts treated with vehicle or IAA. BiP served as a loading control. Representative of two independent replicates. (**C**) Immunofluorescence assays (IFA) analysis of GAP45, CDPK1, PhIL1, and MSP1 expression in schizonts treated with vehicle or IAA. Purified mature schizonts were treated with IAA for 12 hr. Parasites *mAID::dhhc2* and *mAID::dhhc2;phil1::4Myc* used are indicated. PhIL1 is an inner membrane complex (IMC) protein without palmitoylation. MSP1 is a plasma membrane protein. Scale bar = 5 μm. (**D**) IFA analysis of GAP45 and CDPK1 in released *mAID::dhhc2* merozoites treated with vehicle or IAA. Scale bar = 5 μm. (**E**) Solubility assay detected membrane association of GAP45, CDPK1, and MSP1 in the *mAID::dhhc2* schizonts treated with vehicle or IAA. PM protein MSP1 and cytoplasmic protein GAPDH are set as control. Representative of two independent replicates. (**F**) Palmitoylation analysis of GAP45 and CDPK1 ectopically expressed in human cells. Human embryonic kidney 293T cells were cotransfected with plasmids coding for the HA-tagged and human codon-optimized DHHC2 (WT) or its catalytic-deficient mutant (C128A), along with the GAP45 and CDPK1. Flag-tagged mouse CD36 (mCD36-Flag) was also cotransfected and serves as a control of evidenced palmitoylated protein. Tubulin served as a loading control. Representative of two independent replicates. (**G**) Palmitoylation analysis of GAP45 and CDPK1 ectopically expressed in human HEK293T cells treated with or without 2-BP. The cells were cotransfected with a HA-tagged and human codon-optimized DHHC2 and a flag-tagged mouse CD36 (mCD36-Flag). Representative of two independent replicates.

The online version of this article includes the following source data and figure supplement(s) for figure 6:

**Source data 1.** (related to *Figure 6A*) Acyl-RAC method detecting palmitoylation of GAP45 and CDPK1 in the mAID::dhhc2 schizonts treated with vehicle or IAA.

**Source data 2.** (related to *Figure 6B*) Acyl-RAC method detecting palmitoylation of GAP45 and CDPK1 in the dhhc1::mAID schizonts treated with

*Figure 6 continued on next page*

*Figure 6 continued*

vehicle or IAA.

**Source data 3.** (related to *Figure 6E*) Solubility assay detected membrane association of GAP45, CDPK1, and MSP1 in the mAID::dhhc2 schizonts treated with vehicle or IAA.

**Source data 4.** (related to *Figure 6F*) Palmitoylation analysis of GAP45 and CDPK1 ectopically expressed in human cells.

**Source data 5.** (related to *Figure 6G*) Palmitoylation analysis of GAP45 and CDPK1 ectopically expressed in human HEK293T cells treated with or without 2BP.

**Figure supplement 1.** Genotyping of genetically modified parasites.

**Figure supplement 1—source data 1.** (related to *Figure 6—figure supplement 1B*) Diagnostic PCR of *phil1::4Myc; mid::dhhc2* parasite line.

(*Tao et al., 1996*), was independent of *Plasmodium* DHHC2 (*Figure 6F*). The palmitoylation of GAP45 and CDPK1 was also significantly reduced in HEK293T culture treated with inhibitor 2-BP (*Figure 6G*). These results in human cells replicate the observations of DHHC2 activity in the *Plasmodium* schizonts and demonstrate the ability of DHHC2 to palmitoylate GAP45 and CDPK1.

## Residues for palmitoylation in GAP45 and CDPK1

To identify the residue(s) of palmitoylation in GAP45, we used an online software CSS-Palm (csspalm. biocuckoo.org) for prediction, which generated six candidate cysteines (C5, C140, C156, C158, C169, and C172) (*Figure 7A*). To test them, we initially generated four constructs expressing HA-tagged GAP45, each with a single or double cysteine-to-alanine mutations (C5A, C140A, C156A/C158A, and C169A/C172A) (*Figure 7A*, *Figure 7—figure supplement 1A,B*). These constructs were episomally expressed in the schizonts. Only the C140A mutant displayed the IMC localization similar to wild-type (WT) GAP45; other mutants (C5A, C156A/C158A, and C169A/C172A) lost the IMC localization (*Figure 7B*). We also tested the GAP45 C-terminal fragment (30–184 aa) which displayed an IMC localization when fused with a N-terminal GFP (*Figure 7A*, *Figure 7—figure supplement 2A and B*). Substitution of C156A/C158A or C169A/C172A impaired the IMC localization of this fragment while C140A did not (*Figure 7A* and *Figure 7—figure supplement 2A,B*), consistent with the localization for HA-tagged GAP45 in *Figure 7B*. These results suggest that these cysteines (C5, C156 and/or C158, C169 and/or C172) are critical for IMC targeting of GAP45 and might be the residues for modification. Indeed, the Acyl-RAC assay detected significantly decreased palmitoylation of GAP45 in the C5A, C156A/C158A, and C169A/C172A mutants, but not the C140A mutant. Thus, for GAP45, there is an association between IMC localization and palmitoylation. Additionally, the degree of palmitoylation was further reduced in the triple cysteine mutants (C5A/C156A/C158A and C5A/C169A/C172A) relative to the single and double mutants (*Figure 7C*).

In CDPK1, two cysteines (C3 and C252) were predicted as the potential residues for palmitoylation (*Figure 7D* and *Figure 7—figure supplement 1C,D*). Using the same approach, we found that only the C3A mutation caused a complete loss in both protein palmitoylation and IMC targeting of CDPK1 in the schizonts while the C252A mutation had no effect (*Figure 7E–F*), suggesting C3 as the critical residue for protein palmitoylation and IMC targeting of CDPK1 in schizonts. Interestingly, the cysteine residues C5, C156, C158, C169, and C172 of GAP45 and C3 of CDPK1 are evolutionarily conserved among different *Plasmodium* species (*Figure 7—figure supplement 1A,C*). Together, these results suggest that C5, C156, C158, C169, and C172 of GAP45 and C3 of CDPK1 are residues for palmitoylation which direct IMC targeting of the proteins in schizonts.

## Palmitoylation in GAP45 and CDPK1 is essential for parasite viability

Lastly, we asked whether the palmitoylation in GAP45 and CDPK1 is essential for protein function and thus parasite viability. The above cysteine to alanine mutation experiments indicated that the palmitoylation of the N-terminal cysteine (C5 in GAP45 and C3 in CDPK1) is required for the correct IMC targeting of proteins. We attempted to replace the C5 with alanine in the endogenous GAP45 of 17XNL parasite using the CRISPR-Cas9 method. A 742 bp DNA donor template containing the nucleotide substitution was used for homologous replacement. Seven sgRNAs were designed for guiding the Cas9 complex to the target DNA. After three independent transfections with each of these seven Cas9/sgRNA plasmids, we failed to obtain the GAP45 C5A mutant parasites. In contrast, a control mutant parasite clone GAP45 C5C was generated with a silent mutation still encoding C5

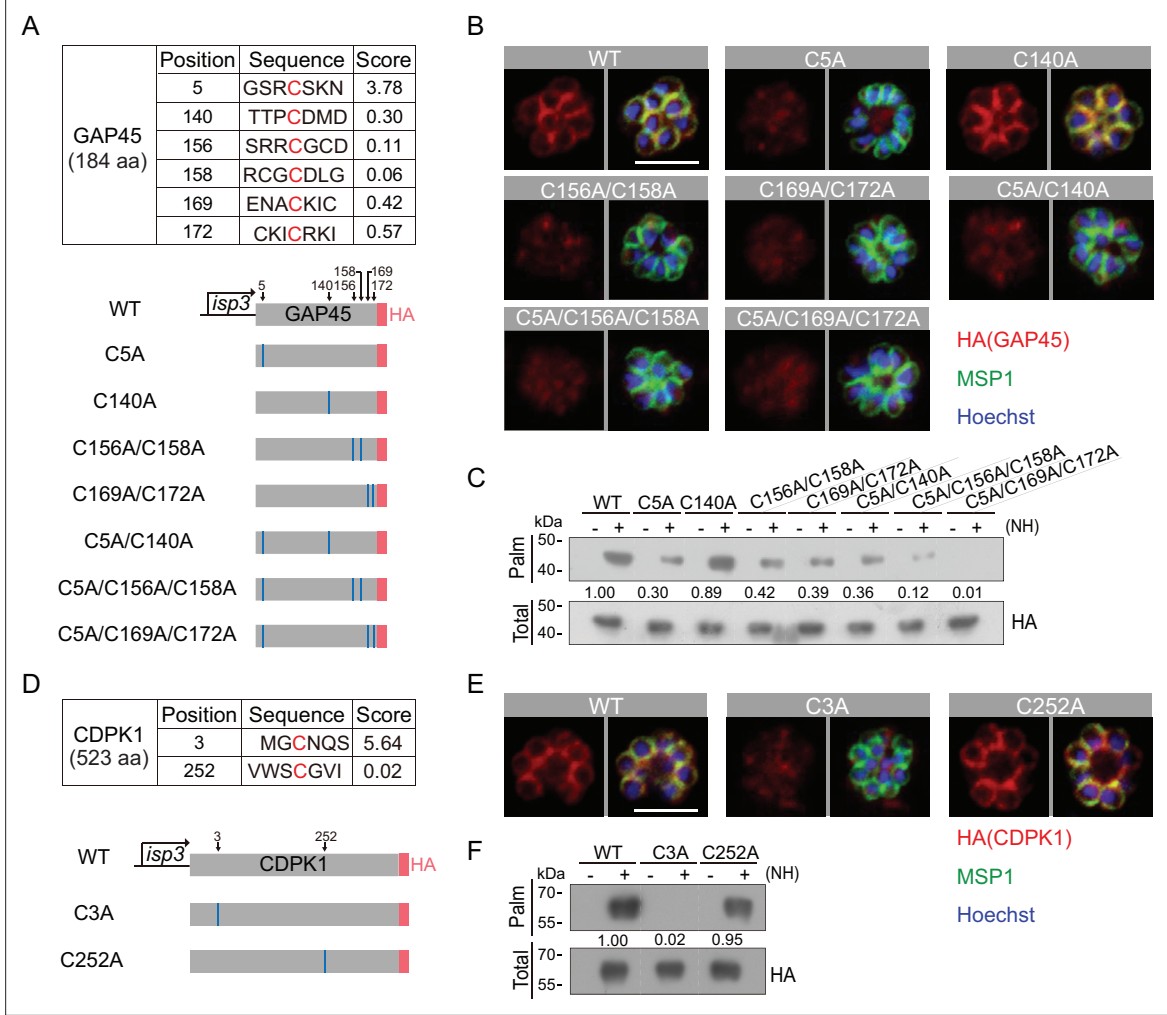

**Figure 7.** Residues for palmitoylation in GAP45 and CDPK1. (**A**) Six cysteine predicated for palmitoylation in GAP45 (upper panel). The score of the predicted palmitoylation sites was provided by the CSS-Palm software. Lower panels indicate schematic of constructs expressing HA-tagged GAP45, each with a cysteine to alanine replacement in single, double, or triple residues (C5A, C140A, C156A/C158A, C169A/C172A, C5A/C140A, C5A/C156A/C158A, and C5A/C169A/C172A). These constructs were episomally expressed in the schizonts. (**B**) Immunofluorescence assays (IFA) of GAP45::HA and seven mutant proteins episomally expressed in schizonts. Scale bar = 5 μm. Representative of three independent repeats. (**C**) Palmitoylation analysis of GAP45::HA and seven mutant proteins episomally expressed in schizonts. Representative of two independent repeats. (**D**) Two cysteine predicated for palmitoylation in CDPK1 (upper panel). The score of the predicted palmitoylation sites was provided by the CSS-Palm software. Lower panels indicate schematic of constructs expressing HA-tagged CDPK1, each with a cysteine to alanine replacement in single residue (C3A and C252A). These constructs were episomally expressed in the schizonts. (**E**) IFA of CDPK1::HA and two mutant proteins episomally expressed in schizonts. Scale bar = 5 μm. Representative of three independent repeats. (**F**) Palmitoylation analysis of CDPK1::HA and two mutant proteins episomally expressed in schizonts. Representative of two independent repeats. HA: hemagglutinin;WT: wild type.

The online version of this article includes the following source data and figure supplement(s) for figure 7:

**Source data 1.** (related to *Figure 7C*) Palmitoylation analysis of GAP45::HA and seven mutant proteins episomally expressed in schizonts.

**Source data 2.** (related to *Figure 7F*) Palmitoylation analysis of CDPK1::HA and two mutant proteins episomally expressed in schizonts.

**Figure supplement 1.** Potential cysteine(s) in GAP45 and CDPK1 for palmitoylation and protein mutants with cysteine replacement to validate the essentiality.

**Figure supplement 2.** Potential cysteine(s) in GAP45 and CDPK1 for palmitoylation and protein mutants with cysteine replacement to validate the essentiality.

(*Figure 7—figure supplement 1E*). Using the same approach, we attempted to replace the C3 with alanine in the endogenous CDPK1 of 17XNL parasite. Similarly, only mutant parasite clones with CDPK1 C3C, but not CDPK1 C3A, were generated (*Figure 7—figure supplement 1F*). Together, these results suggest that palmitoylation of C5 in GAP45 and C3 in CDPK1 is essential for protein function and parasite viability in the asexual blood stage development.

## Discussion

In this study, we attempted a proteome analysis of IMC in the schizonts using the biotin ligase TurboID-mediated PL. Besides abundant proteins relatively easily detected by conventional immunoprecipitation, PL enables detection of proteins with weak or transient interactions. TurboID achieved satisfactory biotinylation in 1 hr, which is much shorter than the 18 hr needed by the BioID. To our knowledge, this is the first application of TurboID-based PL in the *Plasmodium*. We obtained a list of 300 Tb-IMC interacting proteins in the schizonts of *P. yoelii*. Among these proteins, 297 have orthologs in *P. falciparum* (*Supplementary file 1*), suggesting a conserved protein composition of IMC in the schizonts of rodent and human malaria parasites. Although the collection of 300 proteins may contain some false positives, two lines of evidence suggest good reliability of this IMC proteome. Firstly, about 50 IMC or IMC-associated proteins have been identified in *Plasmodium* to date (*Ferreira et al., 2020*; *Wichers et al., 2021*). Among these, 30 proteins (60%) have orthologs included in the Tb-IMC interacting proteins (*Supplementary file 1*). The absence of certain known IMC or IMC-associated proteins in the collection of Tb-IMC interacting proteins may attribute to the following reasons: restricted labeling radius of TurboID, lack of lysines for biotinylation, sterically inaccessibility of proteins by TurboID, or protein expression in other parasite stages (*Sanchez et al., 2021*). Secondly, subcellular localization analysis of 22 candidates confirmed the IMC localization of 8 known and 11 previously undescribed proteins. The localization results strongly suggest a predominantly IMC localization of the proteins tested, although the precise localization at the parasite PM or SPMT cannot be differentiated because of close proximity between IMC and these structures. The exact localization of these proteins at the parasite pellicle needs to be determined in the future using other methods, such as super resolution imaging, immunoelectron microscopy, or split green fluorescent protein. We noted that Wichers et al., recently explored an IMC proteome in asexual blood stages of *P. falciparum* with BioID using PhIL1 as the bait (*Wichers et al., 2021*). Of the 225 PhIL1-interacting protein candidates, the orthologs of 37 proteins are listed among the 300 Tb-IMC interacting proteins in our study (*Supplementary file 1*). We speculate that the differences in protein quantity and depth by PL are probably attributed to the species of biotin ligase, expression abundance of ligase, and the bait strategy.

PL experiments could produce a considerable number of false positives if proper controls are not included for ratiometric or statistical analysis (*Qin et al., 2021*). In a rational design, the PL enzyme is fused to the protein bait of interest while the PL enzyme alone works as a reference control (*Qin et al., 2021*). To achieve the PL of IMC, the TurboID ligase is fused to an IMC targeting peptide (N-terminal 20 aa of ISP1) (*Wang et al., 2020*), directing the IMC targeting of ligase (Tb-IMC) through the ER/Golgi secretary pathway. It is worth noting that the cytoplasm-facing feature of Tb-IMC ligase may prevent the detection of proteins in the IMC lumen. Background levels of ligase activity are set by a ligase not fused to the IMC targeting peptide. This ligase (Tb-cyto), as a spatial reference control, would not enter the ER/Golgi secretory pathway but instead stays in the cytosol or nucleus. Therefore, the Tb-IMC likely measures both interactions with the IMC proteins and interactions with the secretory pathway. Indeed, the 300 Tb-IMC interacting proteins were segregated into two distinct subgroups based on interaction network (*Figure 1E*). Most of known IMC or IMC-associated proteins were clustered into the subgroup I. In contrast, the subgroup II included many ER/Golgi secretory pathway-related proteins. The major role of the subgroup I proteins is likely in IMC function, while the major role of the subgroup II proteins is likely in the secretory pathway. It is possible that some Tb-IMC interacting proteins may have roles in both the IMC and secretory pathway. Detection of the subgroup II proteins in this study provided another independent evidence supporting the IMC formation from the ER/Golgi-derived secretory system. In addition, concomitant capture of subgroup I and II as the Tb-IMC interacting proteins may reflect the tight and dynamic association between the IMC organelle and the endomembrane system in schizont development.

The list of increased IMC proteins allowed us to investigate the localization determinant(s) of IMC proteins which had remained unclear. Several mechanisms underlying IMC targeting have been

proposed, including vesicle-mediated transporting of proteins containing a signal peptide, IMC membrane trapping of proteins with lipid modification, protein motif (alveolin repeats)-mediated IMC localization, and protein-protein interaction for IMC targeting (*Kono et al., 2012*). Palmitoylation is a post-translation modification in which a cysteine residue undergoes a reversible lipid modification, regulating localization and function of target proteins (*Bijlmakers and Marsh, 2003*). Previous studies have implied that palmitoylation mediates the binding or trafficking of proteins to the IMC (*Wang et al., 2020*; *Wetzel et al., 2015*). In *T. gondii,* many glideosome-associated proteins were also palmitoylated (*Foe et al., 2015*). Interestingly, out of the 300 *P. yoelii* Tb-IMC interacting proteins, 83 (28%) have orthologs of the *P. falciparum* palmitoylated proteins in the schizonts (*Jones et al., 2012*). These 83 proteins include GAP45 and ISP3 with validated palmitoylation and CDPK1, GAP50, IMC1g, and IMC1c with predicted palmitoylation. These results imply that a high proportion of IMC proteins are palmitoylated. Palmitoylation of these proteins may be critical for their localization and function. Interaction of these palmitoylated IMC proteins with other non-palmitoylated proteins may also mediate the IMC localization of the latter.

11 putative PATs (DHHC1–11) are encoded in the genomes of rodent malaria parasites while there are 12 PATs (DHHC1-12) in *P. falciparum* (*Hodson et al., 2015*). So far, viable mutant clones were obtained for only DHHC3, DHHC5, DHHC9, and DHHC10 in previous knockout attempts (*Santos et al., 2016*; *Tay et al., 2016*; *Hopp et al., 2016*), suggesting other PATs (DHHC1, DHHC2, DHHC4, DHHC6, DHHC7, DHHC8, DHHC11, and DHHC12) may be essential for the asexual blood stage development. However, the association of these PATs with IMC has not been investigated in the schizonts. In this study, DHHC2 and DHHC1 are among the 300 Tb-IMC interacting proteins and both displayed clear IMC localization in the schizonts. Protein abundance assays and STRING analysis suggested that DHHC2 is likely the main mediator of palmitoylation of the IMC proteins in the schizonts. To determine if DHHC2 contributes to parasite development in vivo, we developed a method for investigating its function in the infected mice. Induced depletion of DHHC2 resulted in a complete growth arrest of the engineered strain *mAID::dhhc2* in mice in an IAA dosage-dependent manner, confirming the essentiality of DHHC2 in the asexual blood stage. In in vitro culture, we observed a significant decrease, but not complete ablation of mature schizont formation in DHHC2-depleted parasites. Most parasites were arrested at a stage earlier than the mature schizont due to defective segmentation or cytokinesis. The defect in asexual blood stage development in vivo and in vitro caused by the loss of DHHC2 was associated with impairment of IMC localization of certain IMC proteins. Indeed, among the 83 potentially palmitoylated IMC proteins, our analysis validated CDPK1 and GAP45 as substrates of DHHC2, but not DHHC1. Aside from CDPK1 and GAP45, other IMC protein substrates palmitoylated by DHHC2 remain to be validated. We previously reported that DHHC2 is critical for parasite development during the zygote-to-ookinete morphogenesis in the mosquito, and DHHC2 palmitoylates ISP1 and ISP3 for their attachment to the IMC to facilitate connection between the IMC and the SPMT (*Wang et al., 2020*). Therefore, DHHC2 plays critical roles in IMC targeting of many proteins at multiple developmental stages.

Compared to the reduction in the *mAID::dhhc2* schizonts caused by IAA treatment, the treatment with 2-BP, a broad-spectrum inhibitor of protein palmitoylation, resulted in no formation of mature *mAID::dhhc2* schizonts (*Figure 6D and E*). The effects of IAA are specific to the mAID-fused DHHC2 in the *mAID::dhhc2* schizonts, whereas 2-BP could inhibit protein palmitoylation mediated by DHHC2 and other PATs, including DHHC1. We suspect that the IMC-residing DHHC1 is also essential for schizont development, but its precise roles and substrate proteins also awaits investigation in the future.

Besides the PATs DHHC2 and DHHC1, several kinases including CDPK1, CDPK4, and PKA, were also found among the Tb-IMC interacting proteins. CDPK1, CDPK4, and PKA were further validated to localize at the IMC of *P. yoelii*, in accordance with their orthologs' localization in *P. falciparum* (*Kumar et al., 2017*; *Fang et al., 2018*; *Wilde et al., 2019*). CDPK1 has been reported to be essential for schizont development. Both conditional knockdown and inhibition of CDPK1 can arrest *P. falciparum* schizont development (*Kumar et al., 2017*; *Azevedo et al., 2013*), mimicking the phenotypes of the DHHC2-depleted parasites in this study. Interestingly, phosphoproteomic analysis revealed that the conditional knockdown of CDPK1 led to the hypophosphorylation of several IMC and glideosome proteins, including GAP45, MTIP, and PKA regulatory subunit (PKAr) (; *Kumar et al., 2017*; *Wilde et al., 2019*; *Green et al., 2008*; *Ridzuan et al., 2012*). In this study, we found that DHHC2-mediated palmitoylation is required for CDPK1 localization at IMC. Thus, palmitoylation and IMC localization are

the prerequisites for CDPK1-mediated phosphorylation of IMC proteins. Based on all available data, we speculated that once the IMC biogenesis is initiated in early schizonts, DHHC2 is recruited to the nascent IMC, possibly by an auto-palmitoylation mechanism (*Wang et al., 2020*). However, DHHC2 seems inessential for IMC biogenesis because the IMC seemed morphologically normal despite DHHC2 depletion (*Figure 5I*). In contrast, DHHC2 palmitoylates certain important proteins for their IMC localization in the developing schizonts and mature merozoites. It has critical function in schizont cytokinesis and erythrocyte invasion.

## Materials and methods
### Plasmid construction and parasite transfection
CRISPR-Cas9 plasmid pYCm was used for parasite genomic modification (*Leung et al., 2017*; *Zhang et al., 2014*). To construct the plasmids for gene deletion, the N- or C-terminal segments (400–600 bp) of the coding regions were PCR-amplified as the left or right homologous arm or 400–600 bp from 5-UTR or 3-UTR following the translation stop codon as left or right arm, respectively. To construct the plasmids for gene tagging, the DNA fragment (encoding 6 HA, 4Myc, GFP, or mAID-2HA) was inserted between the left and right arms in frame with the gene of interest. For each gene tagging, two sgRNAs were designed to target sites close to the N- or C-terminal part of the coding region. To construct the plasmid for amino acid substitution, the donor template (700–800 bp) for homologous recombination was introduced with the targeted mutations for amino acid substitution and extra shield mutations via mutagenesis. These shield mutations in or adjacent to the protospacer-adjacent motif (PAM) were used to prevent the recognition and cleavage of the replaced locus by the gRNA/Cas9 complex. Seven sgRNAs were designed to target sites close to the desired mutation sites. The PCR primers and DNA oligonucleotides used are listed in *Supplementary file 3*. Blood with 15–25% parasitemia was collected from infected mice and cultured in RPMI-1640 (Gibco, cat#11,879,020) supplied with 20% FBS (Gibco, cat#10,099) at 37°C for 3 hr for schizont development. After washing two times with RPMI-1640, the parasite were electroporated with 5–10 µg purified circular plasmid DNA using Lonza Nucleotector. Transfected parasites were immediately intravenously injected into a naïve mouse and applied to pyrimethamine pressure (provided in drinking water at concentration 7 mg/ml) from 24 hr post-transfection. Parasites with transfected plasmids usually appear about 5–7 days after drug selection. Genomic DNA of parasites were extracted for genotyping PCR analysis. Parasite clones with correct modification were obtained using the limiting dilution method.

### Genotypic analysis of transgenic parasites
All transgenic parasites were generated from the *P. yoelii* 17XNL or *Tir1* strains. Infected blood samples from transfected mice were collected from the tail of mice and lysed by 1% saponin in PBS. Parasite genomic DNAs were extracted using DNeasy Blood Kits (Qiagen). For each genetic modification, both 5′ and 3′ homologous recombination were detected by PCR to confirm successful integration of homologous arms. The PCR primers used for genotyping are listed in *Supplementary file 3*. Positive clones with correct modifications were obtained after limiting dilution. The PCR-genotyping results confirming the genetic modified parasites were shown in the corresponding supplementary figure.

### Parasite negative selection with 5-fluorouracil
Modified parasites subjected for sequential modification were negatively selected to remove pYCm plasmid. 5-fluorouracil (5-FC, Sigma-Aldrich, cat#F6627) was prepared in drinking water at a final concentration of 2.0 mg/ml. A naïve mouse receiving parasites with residual plasmid was subjected to 5-FC selection for 6–8 days. Diagnostic PCR was performed to confirm the complete removal of plasmid in the parasites. The PCR primers used are listed in the *Supplementary file 3*.

### DNA mutagenesis for amino acid replacement
For the amino acid replacement in the proteins of interest, the WT gene cDNAs were cloned into NheI and NcoI sites in the pL0019-HA/Myc vector (*isp3* gene promoter for parasite asexual blood stages) or pcDNA3.1-HA/Myc vector (CMV promoter for mammalian cell). A PCR-based protocol with mutagenic oligonucleotides was used to generate the gene mutants. The primers used are listed in *Supplementary file 3*.

## Protein transient expression in asexual blood stage parasites

For protein transient expression, the coding sequence of target genes was tagged N- or C-termi-nally with an epitope tag and driven by the regulatory regions of the *isp3* gene (1.5 kb of the 5′-UTR and 1 kb of the 3′-UTR). Gene expression cassettes were inserted into the pL0019-derived vector containing a human *dhfr* marker for pyrimethamine selection. Blood stage parasites were electropo-rated with 10 µg vector plasmid DNA and selected with pyrimethamine (70 µg/ml) for 7 days. Parasites appearing after pyrimethamine selection were used for further experiments.

## Antibodies and antiserum

The primary antibodies used included: rabbit anti-HA (Western blot, 1:1000, IFA, 1:1000, 3724 S, Cell Signaling Technology [CST]), mouse anti-HA (IFA, 1:500, sc-57,592, Santa Cruz Biotechnology), rabbit anti-Myc (Western blot, 1:1000, 2276 S, CST), mouse anti-Myc (IFA, 1:500, sc-40, Santa Cruz Biotech-nology) and rabbit anti-Histone H3 (Western blot, 1:2000, 9715, CST). The secondary antibodies used included: goat anti-rabbit IgG HRP-conjugated and goat anti-mouse IgG HRP-conjugated secondary antibody (1:5000, Abcam), Alexa Fluor 555 goat anti-rabbit IgG (1:1000, Thermo Fisher Scientific), Alexa Fluor 488 goat anti-rabbit IgG (1:1000, Thermo Fisher Scientific), Alexa Fluor 555 goat anti-mouse IgG (1:1000, Thermo Fisher Scientific), Alexa Fluor 488 goat anti-mouse IgG (1:1000, Thermo Fisher Scien-tific), and Alexa Fluor 488 conjugated streptavidin (1:1000, Invitrogen, S32354). Antiserums, including rabbit anti-GAP45 (Western blot, 1:1000, IFA, 1:1000), mouse anti-GAP45 (IFA, 1:1000), and rabbit anti-BiP (Western blot, 1:1000) were described in previous studies (*Wang et al., 2020*). Other antise-rums, including rabbit anti-CDPK1 (Western blot, 1:1000, IFA, 1:500), rabbit anti-Erd2 (Western blot, 1:1000), and rabbit anti-MSP1 (Western blot, 1:2000, IFA, 1:1000) were prepared by immunization of rabbit or mouse with recombinant protein as antigens: for CDPK1 ($D_{11}$VRGNK…CDNKPF$_{523}$), for Erd2 ($E_{38}$LYLIV…PFNGEV$_{221}$), and for MSP1 ($V_{1413}$YTKRL…GVFCSS$_{1752}$).

## Immunofluorescence assays

Cells were fixed with 4% paraformaldehyde for 15 min and rinsed with PBS three times. The cells were then permeabilized with 0.1% Triton X-100 for 10 min, rinsed with PBS twice, and incubated with 5% BSA for 1 hr. They were incubated with the primary antibodies overnight at 4°C, rinsed with PBS three times, and incubated with fluorescent conjugated secondary antibodies for 1 hr in the dark. After washing with PBS, the cells were stained with DNA dye Hoechst 33342 for 10 min and mounted on glass slides using the mounting medium. Images were captured using identical settings under Zeiss LSM 880 confocal microscope.

## Live cell imaging

Parasites expressing GFP-fused proteins were collected in 200 µl PBS, washed twice with PBS and stained with Hoechst 33342 at room temperature for 10 min. After centrifugation at 300 g for 5 min, the parasites pellets were resuspended in 100 µl of 3% low melting agarose, spread evenly on the bottom of 35 mm plate, and followed by cooling at RT for 15 min. The parasites were imaged by a Zeiss 880 confocal microscope with the 63×/1.40 oil objective.

## Ultrastructure expansion microscopy (U-ExM)

Purified schizonts were sedimented on a 15 mm round poly-D-lysine (Sigma-Aldrich, cat#A-003-M) coated coverslips for 10 min. To add anchors to proteins, coverslips were incubated for 5 hr in 0.7% formaldehyde (FA, Sigma-Aldrich, cat#F8775)/1% acrylamide (AA, Sigma-Aldrich, cat#146,072) at 37°C. Next, gelation was performed in ammonium persulfate (APS, Sigma-Aldrich, cat#A7460)/ N,N,N′,N′-tetramethyl ethylenediamine (Temed, Sigma-Aldrich, cat#110-18-9)/monomer solution (23% sodium acrylate (SA, Sigma-Aldrich, cat#408,220); 10% AA; 0.1% N,N′-methylenebisacryl-amide (BIS-AA, Sigma-Aldrich, cat#M7279) in PBS) for 1 hr at 37°C. Sample denaturation was performed for 60 min at 95°C. Gels were incubated in bulk ddH$_2$O at room temperature overnight for the first expansion. In the following day, gel samples were washed with PBS twice for 30 min each to remove excess of ddH$_2$O. Gels were cut into square pieces (~1 cm×1 cm), incubated with primary antibodies at 37°C for 3 hr, and washed with 0.1% PBS-Tween (PBS-T) three times for 10 min each. Incubation with the secondary antibodies was performed for 3 hr at 37°C followed by three times washes with 0.1% PBS-T for 10 min each. Nuclear was stained by Hoechst 33342. After

the final staining, gels were washed with 0.1% PBS-T three times for 15 min each and expanded overnight by incubating in bulk ddH$_2$O at room temperature. After the second round of expansion, gels were cut into square pieces (~0.5 cm×0.5 cm) and mounted by a coverslip in a fixed position for image acquiring.

## Airyscan super-resolution microscopy

Parasites were imaged using a 100×/1.46 NA oil immersion objective on a Zeiss LSM880 fitted with an Airyscan detector. Super-resolution reconstructions of multi-labeled *dhhc2::6* HA schizonts were acquired, sequentially in three channels, as follows: channel 1=561 nm laser (HA), channel 2=488 nm laser (GAP45), channel 3=405 nm laser (Hoechst 33342). Images were acquired using a defined region of interest (ROI) with an average of two, with 2048×2048 pixels of image size and 8-bit image depth. Every part of each image remains fully within the dynamic range of pixel intensity. Three-dimensional (3D) Z-stacks were acquired at 0.25 µm intervals in Z axis using piezo drive prior to being Airyscan processed in 3D using batch mode in ZEN Black (Zeiss). Maximum intensity projections of super-resolution images were output with the default setting.

## Protein extraction and immunoblot

Parasite total proteins from asexual blood stages were extracted with RIPA Lysis Buffer (50 mM pH 7.4 Tris, 150 mM NaCl, 1% Triton X-100, 1% sodium deoxycholate, 0.1% SDS, 1 mM EDTA) containing protease inhibitor cocktail and PMSF. After ultra-sonication, the lysates were incubated on ice for 30 min before centrifugation at 12,000 g for 10 min at 4°C. The supernatant was then lysed in Laemmli sample buffer, stored at 4°C for immunoblot. Protein samples were separated in SDS-PAGE, transferred to PVDF membrane that was blocked by 5% skim milk in TBST, and then incubated with primary antibodies. After incubation, the membrane was washed three times with TBST and incubated with HRP-conjugated secondary antibodies. The membrane was washed four times in TBST before the enhanced chemiluminescence detection.

## Dot blot detecting biotinylated proteins

Dot blot assay of the biotinylated proteins was performed as described previously (**Magi and Liberatori, 2005**). Freshly extracted proteins were quantified using the Pierce BCA Protein Assay kit (Thermo Scientific, cat#23,227). PVDF membrane preactivated by methanol was prepared. Equal amounts of total proteins (~2 µg) from each sample were loaded onto the PVDF membrane surface. After protein absorbing and air-dry, the PVDF membrane was blocked by 5% skim milk in TBST and incubated with HRP-conjugated streptavidin (GenScript, M00091) for the enhanced chemiluminescence detection. MSP1 protein was used as the loading control.

## Protein solubility assay

Purified schizonts (1×10$^6$) from each sample were used for protein solubility assay. Parasites were lysed in 100 µl of hypotonic buffer (10 mM Hepes, 10 mM KCl, pH 7.4) and frozen and thawed (–80 to 37°C) twice for cell lysis. The lysates were centrifuged at 12,000 g for 5 min at 4°C, and the supernatants containing cytosolic soluble proteins were collected as 'Hypo' fraction. The pellet was then rinsed with 1 ml of ice-cold PBS, resuspended in 100 µl of freshly prepared carbonate buffer (0.1 M Na$_2$CO$_3$), kept on ice for 30 min, and then centrifuged at 12,000 g for 5 min at 4°C. The supernatants containing peripheral membrane proteins were collected as 'Carb' fraction. The pellet was rinsed with 1 ml of ice-cold PBS, resuspended in 100 µl of freshly prepared Triton X-100 buffer (1% Triton X-100), kept on ice for another 30 min, and centrifuged at 12,000 g for 5 min at 4°C. The supernatants containing integral membrane proteins were collected as 'Trx' fraction. The final pellet including insoluble proteins and non-protein materials was solubilized in 1×Laemmli sample buffer as 'pellet' fraction. All fractions were boiled at 95°C for 10 min and centrifuged at 12,000 g for 5 min. Equal volume of supernatants from each sample was used for immunoblot. For detecting the change in IMC localization of palmitoylated proteins, Hypo fractions were referred to 'light fraction', while Hypo-insoluble fractions were referred to 'heavy fraction'. All buffers used in this assay contain the protease inhibitor cocktail (MedChemExpress, cat#HY-K0010).

## Purification of schizont and ring/trophozoite stage parasites

The asexual blood stage parasites with 15–25% parasitemia were cultured in RPMI-1640 medium supplied with 20% fetal bovine serum (FBS) and 100 IU penicillin, 100 mg/ml streptomycin at 37°C for 3 hr for increasing schizont production. The schizonts were purified using the Nycodenz density gradient centrifugation. Briefly, the schizonts were suspended in RPMI-1640 and 7 ml of the schizont culture was loaded on the top of a 2 ml of the 60% Nycodenz solution in a 15 ml centrifugation tube. After centrifugation at 300 g for 30 min and removing the supernatants, top layer containing the schizonts were collected. Rings and early trophozoites were collected at the bottom with uninfected erythrocytes.

## TurboID-based proximity-labeling and pull-down

The schizonts expressing the biotin ligase TurboID or BioID were purified using the methods described above. After incubating with 100 µM biotin (Sigma-Aldrich, cat#B4639) at 37°C for 3 hr, the schizonts were lysed with 0.01% saponin and stored at –80°C. For pull-down, parasites were lysed in RIPA buffer (50 mM Tris-HCl pH 7.4, 150 mM NaCl, 1% NP40, 0.1% SDS, 1% sodium deoxycholate, 1% TritonX-100, 1 mM EDTA) containing protease inhibitor cocktail and PMSF. 10 mg of cell lysates were collected as a biological replicate and incubated with 100 µl of streptavidin sepharose (Thermal Scientific, cat#SA10004). After incubation overnight at 4°C, streptavidin beads were then washed with the following procedures: twice with RIPA lysis buffer, once with 2 M urea in 10 mM Tris-HCl, pH 8.0, and two more times with 50 mM Tris-HCl (pH 8.5). The washed beads were resuspended in 200 µl 50 mM Tris-HCl (pH 8.5) before further trypsin digestion of the biotinylated proteins.

## Protein digestion and peptide desalting

The enriched biotinylated proteins were digested on-bead by rolling with 1 µg of trypsin for 16 hr at 37°C followed by a second digest with 0.5 µg trypsin for 2 hr. For digested peptide samples, StageTips packed with SDB-RPS (2241, 3 M) material (made in-house) were used for desalting. Brifely, about 1% trifluoroacetic acid (TFA; Sigma-Aldrich, cat#T6508) was added into the reactions to stop digestion. The SDB-RPS StageTips were conditioned with 100 µl 100% acetonitrile (ACN) (Sigma-Aldrich, cat#3485). The peptides were loaded into StageTips, followed by centrifugation at 4000 g for 5 min. StageTips were washed twice with 100 µl 1% TFA/isopropyl alcohol (Sigma-Aldrich, cat#I9030), and then washed with 100 µl 0.2% TFA. Elution of peptides was performed using 80% ACN/5% ammonia water. All eluted materials were collected in glass vials (A3511040; CNW Technologies) and dried at 45°C using a SpeedVac centrifuge (Eppendorf Concentrator Plus; 5305).

## Mass spectrometry

Digested peptides were dissolved in 0.1% formic acid (Sigma-Aldrich, cat#06,440) containing independent retention time (iRT) peptides and analyzed by Sequential Window Acquisition of All Theoretical Mass Spectra (SWATH-MS) on TripleTOF 5600. For SWATH-MS, an MS1 scan records a 350–1250 m/z range for 250ms, and a 100–1800 m/z range was recorded for 33.3ms in the high-sensitivity mode MS2 scan. One MS1 scan was followed by 100 MS2 scans, which covered a precursor m/z range from 400 to 1200. SWATH-MS wiff files were converted to centroid mzXML files using MSConvert (version 3.0.19311), which were then subjected to DIA-Umpire software (version 2.1.6) for analysis. Signal-extraction module of DIA-Umpire was used to generate pseudo-DDA mgf files. These mgf files were converted to mzML files, which are subjected to database search using MSFragger (version 2.3) through the FragPipe interface (https://fragpipe.nesvilab.org/). The search parameters were set as followed: precursor monoisotopic mass tolerance '50 ppm' fragment mass tolerance '0.1 Da', modification '57.021464@C', potential modification mass '15.994915@M', cleavage 'semi' and maximum missed cleavage sites '1'. PeptideProphet, ProteinProphet and FDR filtering were performed by Philosopher software (version 3.2.2) (https://github.com/Nesvilab/philosopher; *Qian, 2022*) through the FragPipe interface (https://fragpipe.nesvilab.org/) (*da Veiga Leprevost et al., 2020*). The pep. xml search results were validated and scored using PeptideProphet followed by analysis with ProteinProphet. The precursor ions and proteins were filtered at 1% FDR. The spectral library was generated by using EasyPQP tool (version 0.1.12) which is integrated in the FragPipe software. SWATH-MS files were converted to profile mzXML files. The spectral library based targeted analysis of SWATH-MS was performed using the QuantPipe tool based on the OpenSWATH-PyProphet-Tric workflow

(*Rosenberger et al., 2017*; *Wang et al., 2021*). The results were filtered at 1% global protein FDR. Statistical analysis by Perseus software (version 1.6.10.43) were performed as previously reported (*Tyanova et al., 2016*). Parasite protein intensities were imported into Perseus. Protein abundances were normalized with total intensities of all proteins per run and then log2 transformed. The Pearson correlation analysis, hierarchical clustering, and volcano plots were performed with default settings.

## Chemical treatment of parasite

To evaluate the effects of 2-BP on parasite protein palmitoylation and localization, 2-BP (Sigma-Aldrich, cat#21,604) was added to parasite culture at a final concentration of 100 μM at 37°C for the time indicated in each experiment. To deplete the target parasite proteins in vitro by auxin-induced protein degradation, the parasite culture was added with a final concentration of 1 mM IAA (Sigma-Aldrich, cat#I2886) or vehicle (DMSO 1:1000) at 37°C for 3–12 hr with ambient ~5% $CO_2$ levels. To deplete the target parasite proteins in mice, mice were administered with either IAA or vehicle intra-peritoneally. Each mouse was injected with 0.2 ml of PBS containing IAA (20 mg/ml IAA, 3 mM NaOH, pH 7.4) or vehicle (similar solution without IAA). The usage of this dosage of IAA in mice was referred to a previous study (*Tupper et al., 2010*).

## Quantitative real-time PCR

Purified parasites were prepared for extraction of total RNAs. Following isolation with TRIzol (Invitrogen), total RNA was purified with the RNAeasy Mini Kit (QIAGEN, cat#74,106). cDNA was then obtained with the TransScript Two-Step RT-PCR SuperMix (TransGen Biotech, cat#AT401-01) and checked afterwards for gDNA contaminations via RT-PCR. The Real-time quantitative PCR was performed using SYBR Green Supermix (Bio-Rad, cat#1708882) in the Bio-Rad iCycler iQ system (Bio-Rad, USA). The primers used are listed in *Supplementary file 3*. All runs under the following conditions: 95°C for 20 s followed by 40 cycles of 95°C for 3 s; 60°C for 30 s. The samples were run in triplicate and normalized to *gapdh* using a ΔΔ cycle threshold-based algorithm, to provide arbitrary units representing relative expression levels. Graphpad 8 was used for statistical analysis.

## Flow cytometry analysis

Parasite-infected erythrocytes were collected from mice via tail vein or in vitro culture, washed twice with PBS, and suspended in PBS with Hoechst 33342 (Thermo Fisher Scientific, cat#62,249) for nuclei staining. The parasites were analyzed by flow cytometry in a BD LSR Fortessa flow cytometer. Parasite-infected erythrocytes were gated using the fluorescence signal of 405 nm (Hoechst 33342), while uninfected erythrocytes were used as a control. All data were processed by FlowJo software.

## In vitro trophozoite to schizont development

Purified rings and early trophozoites via Nycodenz centrifugation were incubated with vehicle or IAA and cultured for 12 hr at 37°C at ambient $CO_2$ (~5%). Mature schizonts developed from the early trophozoites were counted by thin blood smears with Giemsa staining.

## Erythrocyte invasion of merozoite

Different from an automatous rupture of mature schizonts of the in vitro cultured *P. falciparum*, the *P. yoelii* schizonts displayed an arrest in rupture after maturation in the in vitro condition. $1.0×10^8$ purified schizonts were incubated with vehicle and IAA, respectively, at 37°C for 3 hr to deplete the DHHC2 protein. After this, the schizonts were mechanically disrupted by vibrating for 1 hr. Under these conditions, merozoite release occurs in about 50% of mature schizonts with both vehicle and IAA treatment. Similar number of merozoites were injected intravenously into each mouse with three to four naïve mice in each group. Mouse blood was collected for blood smears at 20 min after injection and the parasitemia of ring stage was quantified by Giemsa solution staining and flow cytometry.

## Mammalian cell culture and transient transfection

HEK293T cells were maintained in Dulbecco's modified Eagle's medium (DMEM) supplemented with 10% FBS, 100 IU penicillin, 100 mg/ml streptomycin at 37°C in a humidified incubator containing 5% $CO_2$. TurboFect transfection reagent (Thermo Fisher Scientific, cat#R0532) was used for cell

transfection. Total DNA for each plate was adjusted to the same amount by using a relevant empty vector. Transfected cells were harvested at 48 hr after transfection for further analysis.

## Detection of protein palmitoylation

Protein palmitoylation was detected using the Acyl-RAC assay described previously (*Wang et al., 2020*). Schizonts were lysed in DHHC Buffer B (2.5% SDS, 1 mM EDTA, 100 mM HEPES, pH 7.5) containing protease inhibitor cocktail and PMSF and incubated on ice for 30 min. After centrifugation at 12,000 g for 10 min, supernatant was collected and treated with 0.1% methyl methanethiosulfonate (MMTS) at 42 °C for 15 min. MMTS was removed by acetone precipitation followed by washing with 70% acetone three times. Protein samples were solubilized in DHHC Buffer C (1% SDS, 1 mM EDTA, 100 mM HEPES, pH 7.5) and were captured on thiopropyl sepharose 6B (GE Healthcare, 17-0402-01) in the presence of 2 M hydroxylamine or 2 M NaCl (negative control) by agitating for 3 hr at room temperature. Loading controls (Input) were collected before addition of thiopropyl sepharose 6B beads. After five times washing with urea DHHC Buffer (1% SDS, 1 mM EDTA, 100 mM HEPES, 8 M urea, pH 7.5), the captured proteins were eluted from thiopropyl sepharose 6B beads in 60 μl DHHC Buffer C supplemented with 50 mM DTT, and mixed with Laemmli sample buffer for further Western blot analysis.

## Immunoelectron microscopy (Immuno-EM)

Purified schizonts were fixed in 0.1 M phosphate buffer (pH 7.4) containing 2% paraformaldehyde and 0.1% glutaraldehyde. Samples were dehydrated in ethanol and embedded in LR Gold resin (cat#14,381-UC, Electron Microscopy Sciences). Ultrathin sections were blocked with 1% BSA for 10 min at room temperature, incubated with the anti-HA antibody at 4°C overnight, and then incubated with goat anti-rabbit IgG conjugated to gold particles of 15 nm diameter (cat#AB-0295G-Gold, Leading Biology) diluted in 1% BSA. Finally, the sections were fixed with 2.5% glutaraldehyde for 10 min and stained with 1% uranyl acetate. The samples were examined and imaged in a Hitachi HT-7800 electron microscope.

## Transmission electron microscopy

For TEM, purified parasites were prefixed with 2.5% glutaraldehyde in 0.1 M phosphate buffer at 4°C overnight, rinsed three times with PBS, then fixed with 1% osmium acid for 2 hr, and rinsed three times with PBS. Fixed samples were dehydrated with concentration gradient acetone. After embedding and slicing, thin sections were stained with uranyl acetate and lead citrate. All samples were imaged using the HT-7800 electron microscope.

## Bioinformatic analysis and tools

The genomic DNA sequences of target genes were downloaded from PlasmoDB database (*Aurrecoechea et al., 2009*). The sgRNAs of a target gene were designed using EuPaGDT (*Aurrecoechea et al., 2017*). Amino acid sequence alignment was analyzed using MEGA5.0 (*Stecher et al., 2014*). The palmitoylation sites in protein were predicted using CSS-Palm 4.0 (*Ren et al., 2008*). Protein interaction among the Tb-IMC interacting proteins was analyzed with the STRING database (https://string-db.org) (*von Mering et al., 2003*), which includes known and predicted interactions stemming from computational prediction, knowledge transfer between organisms, and interactions aggregated from other databases. Subgroups were divided by k-means clustering option and number of clusters was set to 2. Graph-based clustering of the entire interactome network was generated and visualized in Cytoscape (*Shannon et al., 2003*), and pruned by removing the unconnected nodes in each subgroup. Gene ontology analysis were performed on the PlasmoDB database (*Aurrecoechea et al., 2009*).

## Quantification and statistical analysis

For quantification of protein expression in Western blot, protein band intensity was quantified using Fiji software from three independent experiments. The signals of target proteins were normalized with that of control proteins. For quantification of protein expression in IFA, confocal fluorescence microscopy images were acquired under identical parameters. Fluorescent signals were quantified using Fiji software. More than 30 cells were analyzed in each group. Protein relative expression was

calculated as the signal intensity compared to that of control group. Statistical analysis was performed using GraphPad Software 5.0. Two-tailed Student's $t$-test or Whiney Mann test was used to compare differences between treated groups and their paired controls. n represents the number of parasite cells tested in each group, or experimental replication. The exact value of n was indicated within the figures. p-value in each statistical analysis was also indicated within the figures.

## Acknowledgements

This work was supported by the National Natural Science Foundation of China (32170427, 31970387, 31872214), the Natural Science Foundation of Fujian Province (2021J01028, 2019J05010), and the 111 Project sponsored by the State Bureau of Foreign Experts and Ministry of Education of China (BP2018017).

## Additional information

### Funding

| Funder | Grant reference number | Author |
| --- | --- | --- |
| National Natural Science Foundation of China | 32170427 | Jing Yuan |
| National Natural Science Foundation of China | 31970387 | Jing Yuan |
| National Natural Science Foundation of China | 31872214 | Jing Yuan |

The funders had no role in study design, data collection and interpretation, or the decision to submit the work for publication.

### Author contributions

Pengge Qian, Conceptualization, Formal analysis, Investigation, Methodology, Validation, Writing - original draft; Xu Wang, Formal analysis, Investigation, Methodology, Validation; Chuan-Qi Zhong, Investigation, Methodology; Jiaxu Wang, Investigation, Resources; Mengya Cai, Investigation, Validation; Wang Nguitragool, Writing - review and editing; Jian Li, Formal analysis, Project administration, Resources, Supervision; Huiting Cui, Formal analysis, Funding acquisition, Investigation, Project administration, Resources, Supervision; Jing Yuan, Conceptualization, Funding acquisition, Investigation, Project administration, Resources, Supervision, Writing - original draft, Writing - review and editing

### Author ORCIDs

Jian Li http://orcid.org/0000-0002-6397-2785
Jing Yuan http://orcid.org/0000-0002-8907-9143

### Ethics

All mouse experiments were performed by approved protocols (XMULAC20140004) by the Committee for Care and Use of Laboratory Animals of Xiamen University. The ICR mice (female, 5 to 6 weeks old) were purchased from the Animal Care Center of Xiamen University.

### Decision letter and Author response

Decision letter https://doi.org/10.7554/eLife.77447.sa1
Author response https://doi.org/10.7554/eLife.77447.sa2

## Additional files

### Supplementary files

• Supplementary file 1. List of IMC protein candidates identified in this study.
• Supplementary file 2. List of 188 filtered-out proteins by transcriptional profile.

- Supplementary file 3. Primers and oligonucleotides used in this study.
- Transparent reporting form

## Data availability

The Mass spectrometry proteomic data have been deposited to the ProteomeXchange Consortium via the PRIDE partner repository with the data identifier PXD028193. All other relevant data in this study are submitted as supplementary source files.

The following dataset was generated:

| Author(s) | Year | Dataset title | Dataset URL | Database and Identifier |
|---|---|---|---|---|
| Qian P, Wang X, Zhong C-Q, Wang J, Cai M, Nguitragool W, Li J, Cui H, Yuan J | 2022 | Inner membrane complex proteomics reveals a palmitoylation regulation critical for intraerythrocytic development of malaria parasite | https://www.iprox.cn/page/project.html?id=IPX0003456000 | ProteomeXchange, PXD028193 |

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
