## [Editor Report]

Apicomplexan parasites, including the malaria parasite Plasmodium, possess a characteristic inner membrane complex (IMC), which plays an essential role in maintaining the parasite shape and regulating motility. In this study, the authors used proximity labeling to determine the IMC proteome of erythrocytic stages in the rodent malaria parasite Plasmodium yoelii. They identify the palmitoyl-acyl-transferase DHHC2 as a key enzyme that regulates the localization of IMC proteins through palmitoylation. This work provides new insights into the function of the IMC in the erythrocytic stages of the malaria parasite.

---

## [Decision Letter]

**Decision letter after peer review:**

Thank you for submitting your article "Inner membrane complex proteomics reveals a palmitoylation cascade regulating intraerythrocytic development of malaria parasite" for consideration by *eLife*. Your article has been reviewed by 3 peer reviewers, one of whom is a member of our Board of Reviewing Editors, and the evaluation has been overseen Vivek Malhotra as the Senior Editor. The reviewers have opted to remain anonymous.

All three reviewers agree that this is an interesting and important study and that the work is robust. However, they also raised a number of points, as detailed in their recommendations to the authors, which need to be addressed to clarify some aspects of the work and revise some of the conclusions.

Specifically, the following points must be addressed:

1) Clarify the strategy to select candidates from the TurboID screen.

2) Clarify the localization of CDPK1 (including by improving imaging).

3) Revise the conclusions on the role of DHHC2-mediated palmitoylation of GAP45 and CDPK1 (including by providing additional control of IMC integrity).

4) Address discrepancies in the analysis of DHHC2-deficient or 2-BP treated schizonts (especially regarding MSP1 labelling).

The reviewers also noted that the title is not appropriate and should be modified.

*Reviewer #1 (Recommendations for the authors):*

The study identifies a large number of putative IMC proteins, with only partial overlap with other studies performed in *P. falciparum*. This suggests a potentially high rate of false positives, although 11 out of 14 new proteins were confirmed to be localized in the IMC. The authors should explain how they selected the 14 candidate proteins. This is important to ensure the absence of bias. Do these proteins correspond to the 22 proteins with assigned GO term IMC (line 219)? It is likely that many of the identified proteins correspond to trafficking-related proteins, as discussed in the text, or plasma membrane proteins (such as PMP1) or cytosolic proteins (since the ligase is exposed on the cytosolic face of the IMC). Did the authors consider using purified merozoites instead of schizonts to apply the TurboID proximity labelling? This might allow a more stringent identification of IMC resident proteins.

In addition, the study missed some known IMC proteins, which should be mentioned in the text and discussed further (line 544). For example, how do the authors explain that ISP1 (which was used to target TurboID to the IMC) was not identified in the screen?

In their initial mass spectrometry results the authors identified 488 proteins but discarded 188 based on gene expression profiling. What are these 188 proteins?

The authors should revise the table in Figure 2B.

– PMP1 has been described as a plasma membrane protein in P. berghei (PMID 28851956).

– pPK1 has been shown to be apically localized in P. yoelii merozoites, not in the IMC (PMID 31953169). How do the authors explain this discrepancy?

– PY17X_1139700 is annotated as TMCO1 in PlasmoDB.

– PY17X_1411000 is annotated as PhIL1-interacting candidate 5 (PIC5) and has been shown to be an IMC protein and should be counted with the known IMC proteins (PMID 33830607 and 34324609).

The data in Figure 3 show that 2BP has a marked effect on palmitoylation and IMC localization of 3 proteins (CDPK1, GAP45 and 1411000). However, it is important to control that 2BP does not completely disrupt IMC integrity, for example using PhIL1 as a marker.

In Figure 6C, why is the GAP45 and CDPK1 signal so low after IAA treatment, although input in western blot looks similar to the vehicle control (Figure 6A)?

Same comment in figure 7B and 7E, where it is not clear whether it is the localization of GAP45/CDPK1 that is affected rather than their level of expression.

What is the effect of DHHC1 depletion on parasite growth and schizont segmentation (line 460-468)?

*Reviewer #2 (Recommendations for the authors):*

1. The proximity labelling experiments using the TurboID are properly controlled and analyzed. Out of 14 candidates selected for further localization assay, 11 were found at the periphery of the parasites confirming a good enrichment of the samples. 8 known proteins of the pellicle were also localized, including CDPK1.

For PY17X_1411000 (Figure 2C), a staining around the residual body is visible from which GAP45 is absent. Would it be possible that this protein is localized at the plasma membrane (PM) rather than the IMC? In Green et al., 2008 and Ridzuan et al., 2012, the authors can distinguish between the IMC and PM localization of their proteins of interest in *Plasmodium falciparum*, PfCDPK1, PfGAP45 and PfMTIP, by observing the signal around the residual body. Would it be also valid in Plasmodium yoelii? They actually used that specificity of signal to claim that PfCDPK1 is at the PM while PfMTIP and PfGAP45 are at the IMC.

2. The palmitoylation of GAP45, CDPK1 and PY17X_1411000 in Figure 3 is convincing. However, I wonder why the staining of MSP1 in parasites treated with 2-BP looks normal while in Figure 5E and 5I, the 2-BP impairs schizont development and segmentation. Are the pictures taken after different timing of exposure to the drug?

3. The data concerning the DHHC2-iKD characterization (Figure 5) convincingly demonstrate the essentiality of this enzyme for parasite survival in vivo due to its role during the development and segmentation of the schizonts.

GAP45 and CDPK1 appear clearly cytosolic in merozoites in Figure 6D but the localization of these proteins is difficult to interpret in schizonts. The immunofluorescence presented for the IAA condition in Figure 6C are too weak; the brightness should be increased. Also, it is puzzling to see again that the development of schizonts looks normal in presence of IAA (PhIL1 and MSP1 staining) in contrast to Figure 5F while the treatment is the same (12 hr from the text). The authors should explain why, in this experiment, the "distribution of MSP1 in the DHHC2-deficient schizonts was also unaffected" (line 455, p15).

4. Identification of the palmitoylation sites in GAP45 and CDPK1.

– In Figure 7A and 7D, it would be of interest to have the score of the predicted palmitoylation sites according to CSS-Palm.

– The choice of these proteins for assessing their targeting to the IMC via palmitoylation needs to be further discussed.

Along the text, the authors assume that CDPK1 is at the IMC although all the literature cited mentioned that it is at the PM (Green et al., 2008; Sebastian et al., 2012) like its orthologue TgCDPK3 (McCoy et al., 2012). This point is important for CDPK1 to be a direct substrate of DHHC2. The palmitoylation of CDPK1 at residue C3 and not C252 is clean and convincing but the authors should discuss and maybe revisit the localization of CDPK1 in Plasmodium yoelii, especially in the sentence line 508-510.

GAP45 is known to spend the space between the PM and the IMC being anchored to the PM via its N-terminal acylation (myristoylation on G2 and palmitoylation on C5) and to the IMC via its C-terminus; a topology conserved between Toxoplasma (Frénal et al., 2010) and *Plasmodium falciparum* (Ridzuan et al., 2012) and likely in Plasmodium yoelii. Therefore the C5 site described here is likely at the PM and not at the IMC. However, the investigation of the other sites is very interesting and undoubtedly dependent of DHHC2 but the episomal construct used here is likely not relevant since the C-terminal tag likely already impacts the IMC localization has shown in *T. gondii*. That might explain why the C5A mutant becomes cytosolic, being not anchored in any membrane of the pellicle. A GFP-C-GAP45(aa30-204) construct, as used in Ridzuan et al., 2012, should solve this problem and determine the residues palmitoylated by DHHC2.

*Reviewer #3 (Recommendations for the authors):*

1. While the paper contains a substantial amount of data, it feels like two independent stories that are not as well connected as they could be. The proximity labelling identifies a number of new IMC proteins, many of which are likely palmitoylated. But then rather than probing these deeper, the authors switch to focus on DHHC2 and its role in the trafficking and function of two well-known IMC protein targets. However, this is still a strong paper and an important contribution to the field.

2. Line 22. The authors claim that "19 of the 22 selected targets were confirmed to localize to the IMC". This is somewhat disingenuous as 8 of these were already known IMC proteins. Of the remaining 14 that were chosen for verification (lines 248-252) – were they randomly selected or were certain criteria applied in the selection process? (e.g. predicted palmitoylation, IMC in *T. gondii* [Rab11B], etc). This should be clarified.

3. Regarding cytosolic localization for the control TurboID.

a) Line 141 – ookinetes – "detected cytosolic protein biotinylation". The staining appears both cytoplasmic and nuclear. This would make sense because the mass of the fusion protein is less than the nuclear pore cutoff (~50kDa) and thus it is likely to freely diffuse into the nucleus.

b) Line 163 – this comment also applies to the Tb-cyto – is it cytoplasmic and nuclear or somehow nuclear excluded in the parasites? (Figure 1C and S2E).

4. Figure S4. Regarding N vs C terminal tagging – I am presuming that the N-terminally tagged Rab11A, Rab11B (1359500) were tagged in this fashion to not disrupt C-terminal cysteines. This would be worth mentioning. Also, why is 1359500 not designated Rab11B? (which has previously been shown to be IMC localized in *T. gondii*)

5. Line 330, Figure 4I – Clarify that this is referring to the multiple spots that colocalize. In addition, there appears to be a separate GAP45 spot? This should be discussed and some arrows would be helpful here.

6. Line 505-510 – The authors note that the key palmitoyated sites are conserved among the Plasmodia. Are these sites conserved in *T. gondii*?

7. Figure 6C, line 451 – It is unclear how the loss of IMC localization occurs upon DHHC2 knockdown as there is no control for the IMC in these images.

8. Figure 1E – A more detailed description of precisely how the data was separated into groups I vs II would be helpful.

9. Figure 1D. Line 190. Were any known IMC proteins excluded from transcriptional profiling which reduced the dataset from 488-300 proteins?

10. Figure S1F – the individual points suggest that this is one of the 3 replicate experiments. It would be better to show a bar graph with the mean of 3 replicate experiments, as done for Figure 4C or 5E.

11. Figure S5A – 7 of the 11 immunofluorescence images appear to show no signal? Are many of the DHHCs unable to be tagged? Perhaps confirm tagging by better imaging, western or PCR. (or at least discuss)

[Editors' note: further revisions were suggested prior to acceptance, as described below.]

Thank you for resubmitting your work entitled "Inner membrane complex proteomics reveals a palmitoylation regulation critical for intraerythrocytic development of malaria parasite" for further consideration by *eLife*. Your revised article has been evaluated by Vivek Malhotra (Senior Editor) and a Reviewing Editor.

The manuscript has been improved but there are some remaining issues that need to be addressed, as outlined below:

The authors fully addressed the reviewers comments in their rebuttal but not in the revised manuscript. Many elements provided in the rebuttal could be included in the manuscript to strengthen the conclusions and clarify some points.

Specifically:

1) line 308: "Notably, CDPK1, GAP45, and PIC5 lost their IMC localization and were found in the cytosol or nucleus after 2-BP treatment (Figure 3E)". The signal is too weak to ascertain the cytosol or nucleus localization of the proteins. In addition, in Fig3E, PhIL1 is barely visible. The authors should optimize the brightness/contrast of the images. Alternatively, they may consider using black on white display, which provides better contrast. The authors should mention in the manuscript their interpretation for the lower signal of GAP45/CDPK1, which is due to loss of protein concentration at the IMC.

2) line 480-481: the text indicates treatment with IAA for 12h, which should prevent schizont formation. This is inconsistent with figure 6 where segmented schizonts are visible. This is also not consistent with the rebuttal, where the authors explain that 2BP treatment was applied to schizonts. This must be clarified and timing of treatment should be specified for each experiment shown. The authors should add the timing of 2-BP treatment in the figure legends for Fig3E, Figure 5E and 5I, and Figure 6C.

3) Figure 6C-D: the authors should optimize the brightness/contrast of the images. Alternatively, they may consider using black on white display, which provides better contrast.

4) The authors state in their rebuttal that "For better understanding, we added the information of ISP1 stage expression at the legend of Figure 2A in the revised manuscript." Please check that this information is included in the text.

5) Figure 1 supplt 2 panel A still refers to ISP1 "signal peptide".

6) The supplemental table with primers should be labelled S3 not S2

7) The U-ExM and immune-EM data strengthen the conclusion that CDPK1 is localized at the IMC and should be included in the manuscript. The authors convincingly show that CDKP1 is likely at the IMC rather that at the plasma membrane (PM) using both immuno-EM and U-ExM. These data are a strong addition to the paper since all the previous literature claim that CDPK1 is at the PM and therefore not at a place where it can be palmitoylated by the IMC-embedded DHHC2. The authors should add the immuno-EM and U-ExM data as a supplementary figure.

8) The GFP N-terminal tagging experiments provide important controls and strengthen the conclusion that C156, C158, C169 and C172 are involved in IMC localization via palmitoylation. These data should be included in the manuscript (as a supplemental figure).

9) The arrowhead pointing at the "only GAP45" dot is barely visible in figure 4I.

---

## [Author Response]

Reviewer #1 (Recommendations for the authors):The study identifies a large number of putative IMC proteins, with only partial overlap with other studies performed in *P. falciparum*. This suggests a potentially high rate of false positives, although 11 out of 14 new proteins were confirmed to be localized in the IMC. The authors should explain how they selected the 14 candidate proteins. This is important to ensure the absence of bias. Do these proteins correspond to the 22 proteins with assigned GO term IMC (line 219)? It is likely that many of the identified proteins correspond to trafficking-related proteins, as discussed in the text, or plasma membrane proteins (such as PMP1) or cytosolic proteins (since the ligase is exposed on the cytosolic face of the IMC).

In the initial manuscript of this study, we chosen 22 hits for their protein subcellular localization test. There are 8 protein hits previously validated localizing at IMC in other *Plasmodium* species and 14 hit candidates whose localization at IMC have not been validated in the *Plasmodium*. These 22 hits were chosen basically in a random manner, but they showed various (high, middle, and low) levels in the enrichment ratio detected by the TurboID-mediated proximity labeling (Figure 2A).

Did the authors consider using purified merozoites instead of schizonts to apply the TurboID proximity labelling? This might allow a more stringent identification of IMC resident proteins.

In this study, we used the purified schizont for the proximity labelling (PL) experiments. We agree with the reviewer’s suggestion that using purified merozoites for PL experiments might allow a more stringent identification of IMC proteins. But in our practice, the *P. yoelii* mature schizonts undergo natural rupture in an extremely low efficacy (1-3%) in the in vitro condition. Currently, it is technically difficult to obtain purified merozoites sample without schizont contamination when using the *P. yoelii* parasite model.

In addition, the study missed some known IMC proteins, which should be mentioned in the text and discussed further (line 544). For example, how do the authors explain that ISP1 (which was used to target TurboID to the IMC) was not identified in the screen?

In the section of Discussion, we discussed several possible reasons that some known IMC proteins failed to be detected in our proteomic (line 561-565).

Previous studies have shown that the endogenous ISP1 is not expressed or in a quite low expression level at the asexual blood stage (EMBO J. 2020, PMID: 32395856; Biol Open. 2013, PMID: 24244852). In this study, only the N-terminal 20 residues of ISP1, as an IMC targeting peptide, was used to direct the TurboID ligase localizing at the schizont IMC for PL of IMC proteins. Therefore, it is reasonable that ISP1 could not be identified in the PL. For better understanding, we added the information of ISP1 stage expression at the legend of Figure 2A in the revised manuscript. In contrast, ISP3, the paralogous protein of ISP1 in *Plasmodium* and expressed at the schizont IMC, has been expectedly detected in the Tb-IMC interacting proteins in this study.

In their initial mass spectrometry results the authors identified 488 proteins but discarded 188 based on gene expression profiling. What are these 188 proteins?

For anyone who may be interested in the 188 proteins discarded, we listed them in a new Supplementary Table S2.

The authors should revise the table in Figure 2B.– PMP1 has been described as a plasma membrane protein in P. berghei (PMID 28851956).

Thank the reviewer for correcting this error in our text. We have removed PMP1 in the Figure 2B, and re-claimed the conclusion “9 out of 12 new proteins were confirmed to be localized in the IMC” instead of “11 out of 14 new proteins were confirmed to be localized in the IMC” in the revised manuscript” (line 260-263).

– pPK1 has been shown to be apically localized in P. yoelii merozoites, not in the IMC (PMID 31953169). How do the authors explain this discrepancy?

We revisited the results of pPK1 localization from Ishizaki *et al.* (Parasitol Int, 2020, PMID 31953169). In their study, Myc-tagged pPK1 was shown to be partially co-localized with all three markers tested, including microneme marker AMA1, IMC marker MTIP, and dense granule marker EBL. Moreover, it is unusual that MTIP did not display a typical IMC-localization in the schizonts (see Figure 2D of their publication).

– PY17X_1139700 is annotated as TMCO1 in PlasmoDB.

Changed in the revised Figure 2.

– PY17X_1411000 is annotated as PhIL1-interacting candidate 5 (PIC5) and has been shown to be an IMC protein and should be counted with the known IMC proteins (PMID 33830607 and 34324609).

We agreed with the reviewer’s comment that the *P. falciparum* orthologous (PF1310700/PIC5) of the PY17X_1411000 had been validated to be localizing at IMC of schizonts in a recent study (Wichers *et al.,* Cell Microbiol. 2021. PMID: 33830607), therefore we listed the PY17X_1411000 as the known IMC protein (line 243-245).

The data in Figure 3 show that 2BP has a marked effect on palmitoylation and IMC localization of 3 proteins (CDPK1, GAP45 and 1411000). However, it is important to control that 2BP does not completely disrupt IMC integrity, for example using PhIL1 as a marker.

Thanks for reviewer’s suggestion. We performed experiments and added the PhIL1 as a control showing the IMC integrity under 2-BP treatment in the revised Figure 3E.

In Figure 6C, why is the GAP45 and CDPK1 signal so low after IAA treatment, although input in western blot looks similar to the vehicle control (Figure 6A)?

The GAP45 and CDPK1 proteins were dispersed through cytoplasm and nucleoplasm when detached from the IMC membranes after IAA treatment. These could be the reason that both two proteins looked weaker in IFA signal when compared to the proteins concentrating at IMC in the vehicle control.

Same comment in figure 7B and 7E, where it is not clear whether it is the localization of GAP45/CDPK1 that is affected rather than their level of expression.

GAP45 and CDPK1 were dispersed through cytoplasm and nucleoplasm when detached from the IMC membranes due to loss of or reduced protein palmitoylation. These could be the reason that both two proteins looked weaker in IFA signal when compared to the proteins concentrating at IMC in the control.

What is the effect of DHHC1 depletion on parasite growth and schizont segmentation (line 460-468)?

Our preliminary data showed that DHHC1 is also important for parasite growth in the *P. yoelii*. We are currently working on the detailed defective phenotype analysis in another project.

Reviewer #2 (Recommendations for the authors):1. The proximity labelling experiments using the TurboID are properly controlled and analyzed. Out of 14 candidates selected for further localization assay, 11 were found at the periphery of the parasites confirming a good enrichment of the samples. 8 known proteins of the pellicle were also localized, including CDPK1.For PY17X_1411000 (Figure 2C), a staining around the residual body is visible from which GAP45 is absent. Would it be possible that this protein is localized at the plasma membrane (PM) rather than the IMC? In Green et al., 2008 and Ridzuan et al., 2012, the authors can distinguish between the IMC and PM localization of their proteins of interest in *Plasmodium falciparum*, PfCDPK1, PfGAP45 and PfMTIP, by observing the signal around the residual body. Would it be also valid in Plasmodium yoelii? They actually used that specificity of signal to claim that PfCDPK1 is at the PM while PfMTIP and PfGAP45 are at the IMC.

Two recent studies investigated the *P. falciparum* orthologous (PF1310700/PIC5) of the PY17X_1411000. Wichers *et al.* revealed that the *P. falciparum* PF1310700 was localizing at IMC of schizonts by confocal microscopy (Wichers *et al.,* Cell Microbiol. 2021. PMID: 33830607). In another unpublished work of “Discovery of novel proteins within the *Plasmodium falciparum* inner membrane complex” by Prof. Jeffrey Dvorin’s lab in the Page 56 of abstract book of the Molecular Parasitology Meeting 2020 (see the screenshot below), they performed a co-IP using the IMC protein PfIMC1g (PF3D7_0525800) as bait followed by unbiased mass spectrometry to identify IMC1g-binding partners. The protein of PF3D7_1310700 was identified among the top hits and was further validated to localize to IMC by confocal microscopy. A reciprocal IP using the PF3D7_1310700 protein as bait followed by an immunoblot also confirmed that PfIMC1g interacts with PF3D7_1310700 protein.

These results from ours and others, strongly indicate PY17X_1411000 is an IMC protein. Since this (PY17X_1411000/PIC5) is a validated IMC protein in the *Plasmodium*, we updated our conclusion “9 out of 12 new proteins were confirmed to be localized in the IMC” instead of “11 out of 14 new proteins were confirmed to be localized in the IMC” in the revised manuscript” (line 260-263).

In the studies of Green *et al.* (2008, JBC, PMID:18768477) and Ridzuan *et al.* (2012, PLoS One, PMID:22479457), they indeed showed that PfCDPK1 localizes at cell periphery in schizonts and merozoites by staining with anti-CDPK1 antibody, and that PfCDPK1 is membrane-associated by extraction with different detergents. However, it is hard to distinguish precise localization of PfCDPK1 between plasma membrane (PM) or IMC because of resolution limit based on the evidences presented in those studies. Interestingly, Green *et al.* (2008, JBC, PMID:18768477) provided additional evidences showing that the IMC proteins PfGAP45 and PfMTIP are substrates of PfCDPK1. In addition, Kumar *et al.* (2017, Nat Commun, PMID: 28680058) also revealed that PfCDPK1 interacts and phosphorylates the IMC proteins PfGAP45 and PfIMC1g. Combining the fact of spatial space (approximately 25-30 nm distance) between PM and IMC, we prefer to believe that CDPK1 is an IMC-associated kinase capable of directly phosphorylating GAP45 and MTIP.

To further address reviewer’s concern about IMC localization of CDPK1, additional experiments were performed using two different methods U-ExM and immune-EM. First, we tried the U-ExM method. The physically expanded merozoites of *P. yoelii* were stained with antibodies targeting MSP1 (PM protein, red) and CDPK1 (green) respectively and observed under the Airy-scan model of confocal microscopy. The results showed that most of CDPK1 signals were localized in the inner side of MSP1 signals.

We also tried the immune-EM method. In these tests, we optimized the conditions for immune-EM. Both the *wildtype* and *cdpk1::6HA* schizonts were stained with the 15 nm gold colloidal-labelled antibody. The results showed that CDPK1 (white triangle) was primarily localized in the inner layer, but not the out layer of parasite periphery of the *cdpk1::6HA* schizonts.

Both the results from U-ExM and immune-EM suggested localization of CDPK1 at IMC. However, it is still not easy to conclude the IMC localization of CDPK1 because of the microscopy resolution limit in both experiments.

2. The palmitoylation of GAP45, CDPK1 and PY17X_1411000 in Figure 3 is convincing. However, I wonder why the staining of MSP1 in parasites treated with 2-BP looks normal while in Figure 5E and 5I, the 2-BP impairs schizont development and segmentation. Are the pictures taken after different timing of exposure to the drug?

In Figure 3E, we evaluated the effect of 2-BP treatment on the localization of IMC proteins in the schizont. In these experiments, it is the mature schizont parasites which were treated with 2-BP. 2-BP treatment seems to have little effect on IMC structure integrity, but affects the localization of the IMC palmitoylated proteins.

In the Figure 5E and 5I, we evaluated the effect of 2-BP treatment on the parasite development from trophozoite to schizont. In these experiments, it is the ring and trophozoite mixed parasites which were treated with 2-BP. 2-BP treatment completely impairs the trophozoite to schizont development with no mature schizont formation.

3. The data concerning the DHHC2-iKD characterization (Figure 5) convincingly demonstrate the essentiality of this enzyme for parasite survival in vivo due to its role during the development and segmentation of the schizonts.GAP45 and CDPK1 appear clearly cytosolic in merozoites in Figure 6D but the localization of these proteins is difficult to interpret in schizonts. The immunofluorescence presented for the IAA condition in Figure 6C are too weak; the brightness should be increased.

The GAP45 and CDPK1 proteins were dispersed through cytoplasm and nucleoplasm when detached from the IMC membranes after IAA treatment. These could be the reason that both proteins looked weaker in IFA signal when compared to the proteins concentrating at IMC in the vehicle control. Since it is not appropriate to increase the brightness of the pictures, we prefer to keep them unchanged.

Also, it is puzzling to see again that the development of schizonts looks normal in presence of IAA (PhIL1 and MSP1 staining) in contrast to Figure 5F while the treatment is the same (12 hr from the text). The authors should explain why, in this experiment, the "distribution of MSP1 in the DHHC2-deficient schizonts was also unaffected" (line 455, p15).

In Figure 6C, we evaluated the effect of DHHC2 depletion by IAA on the localization of IMC proteins in the schizont. In these experiments, it is the mature schizont parasites which were treated with IAA. As expected, IAA treatment impaired the localization of the palmitoylated IMC proteins GAP45 and CDPK1, but had little effect on the non-palmitoylated IMC protein PhIL1 or the PM protein MSP1.

4. Identification of the palmitoylation sites in GAP45 and CDPK1.– In Figure 7A and 7D, it would be of interest to have the score of the predicted palmitoylation sites according to CSS-Palm.

Thanks for reviewer’s suggestion. We added the score in the revised Figure 7A and 7D.

– The choice of these proteins for assessing their targeting to the IMC via palmitoylation needs to be further discussed.Along the text, the authors assume that CDPK1 is at the IMC although all the literature cited mentioned that it is at the PM (Green et al., 2008; Sebastian et al., 2012) like its orthologue TgCDPK3 (McCoy et al., 2012). This point is important for CDPK1 to be a direct substrate of DHHC2. The palmitoylation of CDPK1 at residue C3 and not C252 is clean and convincing but the authors should discuss and maybe revisit the localization of CDPK1 in Plasmodium yoelii, especially in the sentence line 508-510.

We revisited the results of CDPK1 localization from Green *et al.* (2008, JBC, PMID:18768477). Green *et al.* attempted to distinguish CDPK1 localization at PM or IMC by co-staining with MTIP and MSP1. The results showed that CDPK1 co-localized with MTIP and MSP1 (see Figure 2B of their publication). It’s hard to conclude precise localization of CDPK1 at PM or IMC based on the images presented in their studies because of resolution limit.

We also revisited the results of Sebastian *et al.* (2012. Cell Host & Microbe, PMID: 22817984). Sebastian *et al.* tagged the endogenous CDPK1 and found that CDPK1 was localized in the peripheral of the schizont and ookinete (see Figure 1C of their publication). However, no precise localization of CDPK1 at PM or IMC was investigated in their studies.

GAP45 is known to spend the space between the PM and the IMC being anchored to the PM via its N-terminal acylation (myristoylation on G2 and palmitoylation on C5) and to the IMC via its C-terminus; a topology conserved between Toxoplasma (Frénal et al., 2010) and *Plasmodium falciparum* (Ridzuan et al., 2012) and likely in Plasmodium yoelii. Therefore the C5 site described here is likely at the PM and not at the IMC. However, the investigation of the other sites is very interesting and undoubtedly dependent of DHHC2 but the episomal construct used here is likely not relevant since the C-terminal tag likely already impacts the IMC localization has shown in *T. gondii*. That might explain why the C5A mutant becomes cytosolic, being not anchored in any membrane of the pellicle. A GFP-C-GAP45(aa30-204) construct, as used in Ridzuan et al., 2012, should solve this problem and determine the residues palmitoylated by DHHC2.

In the reviewer’s concern, the C-terminal tag may likely impact the IMC localization of GAP45 according to the observation in *T. gondii*. In Figure 7B，both GAP45-WT and GAP45-C140A were tagged with a HA epitope, similarly as GAP45-C5A and other mutants. GAP45-WT and GAP45-C140A displayed IMC localization while GAP45-C5A and other mutants were in the cytosol under same condition. These results partially exclude the deleterious effect of the C-terminal HA tag on the IMC localization of GAP45 in the schizonts of *P. yoelii.*

To further address reviewer’s concern, we followed the reviewer’ suggestion and analyzed the GAP45(aa30-184) tagged with an N-terminal GFP. Note the protein length of GAP45 is 184 aa in *P. yoelii* and 204 aa in *P. falciparum*. Again, we observed the IMC localization of GFP-GAP45(30-184). Furthermore, the C140A mutation seemed to have little effect on the IMC localization of GFP-GAP45(30-184) while either C156A/C158A or C169A/C172A mutation impaired the IMC localization of GFP-GAP45(30-184), which are consistent with the results of the HA-tagged GAP45 mutants shown in the Figure 7B. Author response image 1 shows the results (not included in the revised manuscript).

**Author response image 1. sa2fig1:** 

Reviewer #3 (Recommendations for the authors):1. While the paper contains a substantial amount of data, it feels like two independent stories that are not as well connected as they could be. The proximity labelling identifies a number of new IMC proteins, many of which are likely palmitoylated. But then rather than probing these deeper, the authors switch to focus on DHHC2 and its role in the trafficking and function of two well-known IMC protein targets. However, this is still a strong paper and an important contribution to the field.

Thanks for the reviewer’s nice comments.

2. Line 22. The authors claim that "19 of the 22 selected targets were confirmed to localize to the IMC". This is somewhat disingenuous as 8 of these were already known IMC proteins. Of the remaining 14 that were chosen for verification (lines 248-252) – were they randomly selected or were certain criteria applied in the selection process? (e.g. predicted palmitoylation, IMC in *T. gondii* [Rab11B], etc). This should be clarified.

In the initial manuscript, we chosen 22 hits for their protein subcellular localization test. There are 8 protein hits previously validated localizing at IMC in other *Plasmodium* species and 14 hit candidates whose localization at IMC have not been validated in the *Plasmodium*. These hits were chosen basically in a random manner, but they showed various (high, middle, and low) levels at the enrichment ratio in the PL experiments (Figure 2A). We added these descriptions in the revised manuscript.

3. Regarding cytosolic localization for the control TurboID.a) Line 141 – ookinetes – "detected cytosolic protein biotinylation". The staining appears both cytoplasmic and nuclear. This would make sense because the mass of the fusion protein is less than the nuclear pore cutoff (~50kDa) and thus it is likely to freely diffuse into the nucleus.

Thanks reviewer for pointing it out. We changed the “cytosolic protein biotinylation in the TurboID-ookinetes” to “cytosolic and nuclear protein biotinylation in the TurboID-ookinetes”.

b) Line 163 – this comment also applies to the Tb-cyto – is it cytoplasmic and nuclear or somehow nuclear excluded in the parasites? (Figure 1C and S2E).

We changed the “The biotinylated proteins and the ligase displayed cytosolic distribution in the Tb-cyto schizonts” to “The biotinylated proteins and the ligase displayed cytosolic and nuclear distribution in the Tb-cyto schizonts”.

4. Figure S4. Regarding N vs C terminal tagging – I am presuming that the N-terminally tagged Rab11A, Rab11B (1359500) were tagged in this fashion to not disrupt C-terminal cysteines. This would be worth mentioning. Also, why is 1359500 not designated Rab11B? (which has previously been shown to be IMC localized in *T. gondii*)

We updated the PY17X_1359500 with Rab11b, and have added the information of N-terminal tagging for the Rab11a and Rab11b (PY17X_1359500) in the figure legend of Figure S4 of the revised manuscript.

5. Line 330, Figure 4I – Clarify that this is referring to the multiple spots that colocalize. In addition, there appears to be a separate GAP45 spot? This should be discussed and some arrows would be helpful here.

We changed the “both proteins were found as separate dots at the periphery of the intact nucleus in the early schizonts” to “both proteins were found as separate dots that co-localize at the periphery of the intact nucleus in the early schizonts” (line 343-344). In the revised Figure 4I, we labeled an arrow for the sporadic “separate GAP45 spot”. This “only-GAP45” dot could be attributed by the difference in protein stain efficiency for DHHC2 and GAP45. In Figure 4I legend, we added this notion. To further address the reviewer’s concern, Author response image 2 presents images of two other early schizonts, which also showed various stain of DHHC2 and GAP45 (not included in the revised manuscript).

6. Line 505-510 – The authors note that the key palmitoyated sites are conserved among the Plasmodia. Are these sites conserved in *T. gondii*?

We performed the protein sequence alignment (see Author response image 3) and found that the 6 cysteine sites in GAP45 are conserved among the *Plasmodian* and *T. gondii*. Only the C3 in CDPK1 is conserved among the *Plasmodian* and *T. gondii*.

**Author response image 3. sa2fig3:** 

7. Figure 6C, line 451 – It is unclear how the loss of IMC localization occurs upon DHHC2 knockdown as there is no control for the IMC in these images.

In this experiment, the PhIL1, an IMC protein without palmitoylation modification, was used as a negative control for tracking the IMC in both vehicle and IAA-treated groups. We have added this information in the figure legend of Figure 6C.

8. Figure 1E – A more detailed description of precisely how the data was separated into groups I vs II would be helpful.

Thanks for reviewer’s suggestion. We added the detailed information in the section of Material and Method (line 1,252-1,259).

9. Figure 1D. Line 190. Were any known IMC proteins excluded from transcriptional profiling which reduced the dataset from 488-300 proteins?

None of the known IMC proteins was excluded by the transcriptional profiling. All the known IMC proteins were included in the Group I representing the IMC or IMC-associated proteins.

10. Figure S1F – the individual points suggest that this is one of the 3 replicate experiments. It would be better to show a bar graph with the mean of 3 replicate experiments, as done for Figure 4C or 5E.

We changed the spot graph to a bar graph with the mean of three replicate experiments in the revised Figure S1F.

11. Figure S5A – 7 of the 11 immunofluorescence images appear to show no signal? Are many of the DHHCs unable to be tagged? Perhaps confirm tagging by better imaging, western or PCR. (or at least discuss)

In our previous work (EMBO J. 2020, PMID: 24244852), we tagged each of the 11 *P. yoelii* endogenous DHHCs (DHHC1-11) with 6HA and successfully obtained the modified strains using the CRISPR-Cas9 method. We had the genotyping evidences confirming correct modification of these strains in that paper. In the Figure S5A, only DHHC1, DHHC2, DHHC7, and DHHC8 were expressed in asexual blood stages while other DHHC proteins were not detected or showed quite low expression level under same condition.

[Editors' note: further revisions were suggested prior to acceptance, as described below.]

The manuscript has been improved but there are some remaining issues that need to be addressed, as outlined below:The authors fully addressed the reviewers comments in their rebuttal but not in the revised manuscript. Many elements provided in the rebuttal could be included in the manuscript to strengthen the conclusions and clarify some points.Specifically:1) line 308: "Notably, CDPK1, GAP45, and PIC5 lost their IMC localization and were found in the cytosol or nucleus after 2-BP treatment (Figure 3E)". The signal is too weak to ascertain the cytosol or nucleus localization of the proteins. In addition, in Fig3E, PhIL1 is barely visible. The authors should optimize the brightness/contrast of the images. Alternatively, they may consider using black on white display, which provides better contrast. The authors should mention in the manuscript their interpretation for the lower signal of GAP45/CDPK1, which is due to loss of protein concentration at the IMC.

Thanks for reviewer’s suggestion. We have changed accordingly in Figure 3E and used the black on white display. In addition, we added the interpretation for lower signal in the revised manuscript (lines 300-301)

2) line 480-481: the text indicates treatment with IAA for 12h, which should prevent schizont formation. This is inconsistent with figure 6 where segmented schizonts are visible. This is also not consistent with the rebuttal, where the authors explain that 2BP treatment was applied to schizonts. This must be clarified and timing of treatment should be specified for each experiment shown. The authors should add the timing of 2-BP treatment in the figure legends for Fig3E, Figure 5E and 5I, and Figure 6C.

We have added the information about timing for 2-BP treatment in the legends for Fig3E, Figure 5E and 5I, and Figure 6C in the revised manuscript.

3) Figure 6C-D: the authors should optimize the brightness/contrast of the images. Alternatively, they may consider using black on white display, which provides better contrast.

Thanks for reviewer’s suggestion. We have changed accordingly in Figure 6C-D and used the black on white display.

4) The authors state in their rebuttal that "For better understanding, we added the information of ISP1 stage expression at the legend of Figure 2A in the revised manuscript." Please check that this information is included in the text.

The information of ISP1 stage expression was provided in Figure 1—figure supplement 2.

5) Figure 1 supplt 2 panel A still refers to ISP1 "signal peptide".

Thanks for pointing it out. We changed the “signal peptide” to the “IMC targeting peptide” in Figure 1—figure supplement 2A.

6) The supplemental table with primers should be labelled S3 not S2

Changed.

7) The U-ExM and immune-EM data strengthen the conclusion that CDPK1 is localized at the IMC and should be included in the manuscript. The authors convincingly show that CDKP1 is likely at the IMC rather that at the plasma membrane (PM) using both immuno-EM and U-ExM. These data are a strong addition to the paper since all the previous literature claim that CDPK1 is at the PM and therefore not at a place where it can be palmitoylated by the IMC-embedded DHHC2. The authors should add the immuno-EM and U-ExM data as a supplementary figure.

Thanks for reviewer’s suggestions. We added both the immuno-EM and U-ExM results in the revised manuscript (lines 260-262) and in the new Figure 2—figure supplement 3A and B.

8) The GFP N-terminal tagging experiments provide important controls and strengthen the conclusion that C156, C158, C169 and C172 are involved in IMC localization via palmitoylation. These data should be included in the manuscript (as a supplemental figure).

Thanks for reviewer’s suggestions. We added these results in the revised manuscript (lines 514-519) and in the new Figure 7—figure supplement 2A and B.

9) The arrowhead pointing at the "only GAP45" dot is barely visible in figure 4I.

We changed arrowhead in size and color in Figure 4I.